**TECHNIQUES AND RESOURCES**

**SPECIAL ISSUE**
**LIFELONG DEVELOPMENT**

# Single-cell transcriptomic profiling of the whole colony of *Botrylloides diegensis*: insights into tissue specialization and blastogenesis

Berivan Temiz[1], Michael Meier[1,2] and Megan J. Wilson[1,*]

## ABSTRACT

*Botrylloides diegensis* is a colonial ascidian that has been the focus of developmental, evolutionary and regeneration research. In this study, we performed single-cell RNA sequencing (scRNA-seq) of an entire *B. diegensis* colony, including zooids, buds and vascular tunics, to resolve cellular heterogeneity and to identify cell and tissue markers. We identified 29 major cell clusters within the colony and used *in situ* hybridization to examine the spatial expression of cluster marker genes. Numerous tissue types were identified at the molecular level, including blood cells and zooid tissues, such as the branchial epithelium, stomach and endostyle. Distinct cluster markers were identified for specific regions of the stomach epithelium, highlighting the specialization of these regions and the strength of using scRNA-seq to explore their functionality. Cell trajectory projections highlighted the early appearance of progenitor clusters, whereas more differentiated zooid-related tissues appeared later in the developmental path. This study provides a valuable resource for understanding the development, tissue function and regeneration of *B. diegensis.* It demonstrates the power of scRNA-seq to define cell types and tissues in complex colonial organisms.

KEY WORDS: Ascidian, Progenitor, Single-cell sequencing, *Botrylloides*

## INTRODUCTION

Single-cell RNA sequencing (scRNA-seq) can quantify the total transcriptome of individual cells in the tissue/organism of interest. This information can be used to classify, characterize and distinguish each cell at the mRNA level, allowing the identification of different cell types, developmental states and trajectories. Many studies have mapped the transcriptomic composition of diverse animals, including invertebrates such as planaria, sponges, *Hydra* and ascidian species (Fincher et al., 2018; Plass et al., 2018; Cao et al., 2019; Zhang et al., 2020). These studies are essential to understanding how cell types differ from each other through the

[1]Developmental Biology and Genomics Laboratory, Department of Anatomy, Otago School of Biomedical Sciences, University of Otago, PO Box 56, Dunedin 9054, New Zealand. [2]Department of Pathology, University of Otago, Dunedin 9054, New Zealand.

*Author for correspondence (meganj.wilson@otago.ac.nz)

B.T., 0000-0002-0499-8301; M.J.W., 0000-0003-3425-5071

activation or repression of specific pathways; therefore, they can explain how cells possess pluripotency, commit to becoming specific biological units, access categorical morphology and become a part of tissue during development or regeneration.

Colonial ascidians are sessile marine chordates (Fig. 1A), which can be categorized into three main parts: (1) zooids, buds and budlets as the asexually developing body; (2) the vascular network, including vessels, blood cells and vascular termini/ampullae; and (3) tunic, which is the gelatinous matrix covering all colonies (Berrill, 1947; Blanchoud et al., 2017) (Fig. 1B,C). The zooid, a filter-feeding individual, comprises key anatomical structures, including the atrial and branchial epithelium, siphons, endostyle, neural complex, digestive organs (stomach, intestine and pyloric gland), gonads, peribranchial sacs and pericardium (Fig. 1D,E) (Berrill, 1947; Kawamura and Sunanaga, 2010; Holland, 2016; Blanchoud et al., 2017; Anselmi et al., 2022).

Blood circulates throughout the colony via a network of vessels embedded in the tunic, connecting with the zooid and bud sinuses. Each zooid has its a heart-driving circulation within interconnected vessels (Burighel and Brunetti, 1971; Mukai et al., 1978). At least 11 blood cell categories have been identified in the hemolymph and are grouped into five cell types: (1) undifferentiated cells, hemoblasts and differentiating cells; (2) immunocytes, hyaline amebocytes, macrophage-like cells, granular amebocytes and morula cells; (3) transport cells, compartment amebocytes and compartment cells; (4) mast cell-like cells or granular cells; and (5) storage cells, pigment cells or nephrocytes (Wright and Ermak, 1982; Cima et al., 2001; Hirose et al., 2003; Blanchoud et al., 2017).

During asexual reproduction, a new zooid is produced by budding from the atrial epithelium of the parental zooid (Berrill, 1947; Mukai et al., 1978; Manni and Burighel, 2006). This process involves the formation of buds, which develop into mature zooids and smaller budlets, representing earlier stages of development. It is hypothesized that budding begins with the dedifferentiation of parental epidermal cells, followed by the formation of internal organs through the differentiation of pluripotent cells derived from the budlet epithelium (Manni and Burighel, 2006). This budding process is morphologically similar to whole-body regeneration (WBR) (Berrill, 1951; Rinkevich et al., 2007). Thus, studying intact whole colonies will help capture the critical aspects of their development, tissue function and regeneration.

Several scRNA-seq studies have been performed with the solidary ascidian *Ciona* species, mainly during the embryonic stage, showing the conservation of developmental and functional programs in chordate evolution (Horie et al., 2018; Cao et al., 2019; Sharma et al., 2019; Zhang et al., 2020). *Botrylloides diegensis*, previously identified as *Botrylloides leachii* in earlier studies (Temiz et al., 2023), is an emerging model for development, regeneration and stem cell

**DEVELOPMENT**

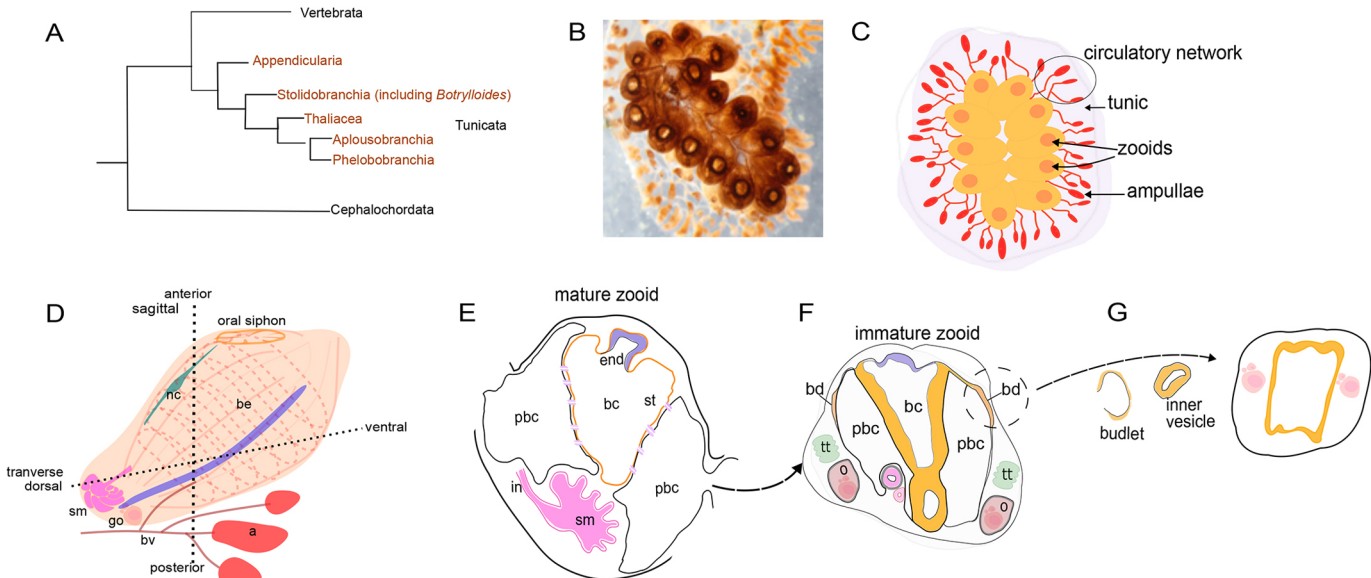

**Fig. 1. The study system: *Botrylloides diegensis*.** (A) Simplified phylogeny showing the position of tunicates (Orders in orange) as the closest chordate group to vertebrates (based on Kocot et al., 2018). (B) A colony of *B. diegensis* attached to a glass slide. (C) Illustration of *B. diegensis* orange morphs with vascular networks (red lines) and elliptical vascular termini (ampulla). Zooids are arranged side-by-side in a ladder-like configuration within the gelatinous tunic. (D) Simplified external anatomy of a mature zooid in a lateral view. A vascular network [blood vessels (bv) with terminal ampullae (a)] connects the zooids throughout the colony. (E) Example schematic of a section through a mature zooid showing structures that are often present. These include the branchial chamber (bc), endostyle (end), peribranchial chamber (pbc), intestine (in) and stomach (sm). (F,G) Section of a primary bud (F) and developmental stages of a budlet (G). The dashed arrow indicates the progression of budlet development. nc, neural complex; go, gonad; be, branchial epithelium; tt, testis; o, ovary; bd, bud disc.

studies, owing to its small genome size, short replenishment period and excellent example of WBR (Blanchoud et al., 2018). Therefore, investigating the composition of mature colonies is crucial to understanding the cell types and tissues of *B. diegensis*.

This study aimed to develop a protocol and resource for *B. diegensis* to identify cell and tissue markers. Single-cell transcriptomic profiling of a mature (blastogenic stage A) *B. diegensis* colony was performed. Consequently, several cell and tissue types were resolved based on their unique gene expression profiles, and their spatial expression was validated using *in situ* hybridization.

## RESULTS

We performed scRNA-seq using acetic acid-methanol (ACME) tissue dissociation and fixation methods (García-Castro et al., 2021) (Fig. 2A). Cells from a *B. diegensis* colony at stage A of the blastogenic cycle were fixed and dissociated using ACME (Fig. S1A,B). This was followed by FACS sorting to remove cell aggregates and debris (Fig. S1E-L). Chromium Single-cell 3′ (10X Genomics) was used for single-cell barcoding, and the resulting library underwent paired-end sequencing. A single-cell transcriptome library containing approximately 58 million reads was generated. The rate of reads that mapped to unique genes was 73%. In total, 6353 cells were detected, with a mean of 1586 UMIs and 481 genes per cell (Table S1).

Cluster analysis was performed to group single-cell transcriptomes based on similarity using the Seurat software. Genes showing high cell variation were calculated to obtain the best signal in cluster differences, and 2000 genes were included by default. Twenty-nine distinct clusters were identified using nonlinear dimensional reduction with Uniform Manifold Approximation and Projection (UMAP) (Fig. 2B). The cell numbers for each cluster are shown in Fig. S4. The highest cell number was observed for cluster 0, with 779 cells, and the lowest cell number was 25 for cluster 28. The mean cell number/cluster ratio was 208. The average number of genes in each

cluster was 291. Differential expression analysis identified the top genes that were potential markers for each cluster (Fig. 2C). Gene ontology (GO) analysis was also conducted for each cluster to aid the functional characterization of the clusters (Table S2; Figs S9-S16).

### Using single-cell data to identify regional tissue markers

The expression patterns of the top marker genes of clusters 4, 5 and 14 were determined by *in situ* hybridization (Fig. 3). *Ctrb1* (*g03753*) was one of the most highly expressed transcripts (~8 LFC) in cluster 4, which was exclusively present in this cluster compared to other clusters (Fig. 3A). *Ctrb1* encodes a serine protease enzyme linked to the acinar-like exocrine glandular cells involved in digestion (Perillo et al., 2016). A probe was designed to determine the cell-type expression of *Ctrb1*, and a strong staining signal was detected in the stomach of the zooid (Fig. 3A). These *Ctrb1*$^+$ epithelial cells were a subset of cells located in the outer curling of the stomach folds (Fig. 3A). No staining was observed in other tissues or vascular cells in mature colonies (Fig. 3A).

*Kng1* (*g08491*) was identified as a marker for cluster 14 (Table S1) and present to a lesser extent in clusters 2, 4 and 5. A 610 bp fragment of *Kng1* was cloned and sequenced for use during *in situ* hybridization. Intense *Kng1* staining was observed within the lateral edges of the stomach via *in situ* hybridization in adult tissue sections (Fig. 3B). Thus, cluster 14 was identified as a part of the stomach epithelium.

The cluster 10 marker *g04846*, a cornifelin-like gene (*Cnfn*), showed specific expression in stigmata cells (Fig. 3C). This gene is localized to microtubules in humans, particularly in the epidermis and oral mucosa (Wagner et al., 2019), and is associated with cell-cell adhesion. It contains a cysteine-rich domain known as a PLAC8 domain.

*Tubb* was upregulated in cells found in the epithelial tissues of the digestive tract, endostyle and stigmata (Fig. 3D). Tubb is a tubulin beta protein that functions in the microtubules of the cytoskeleton,

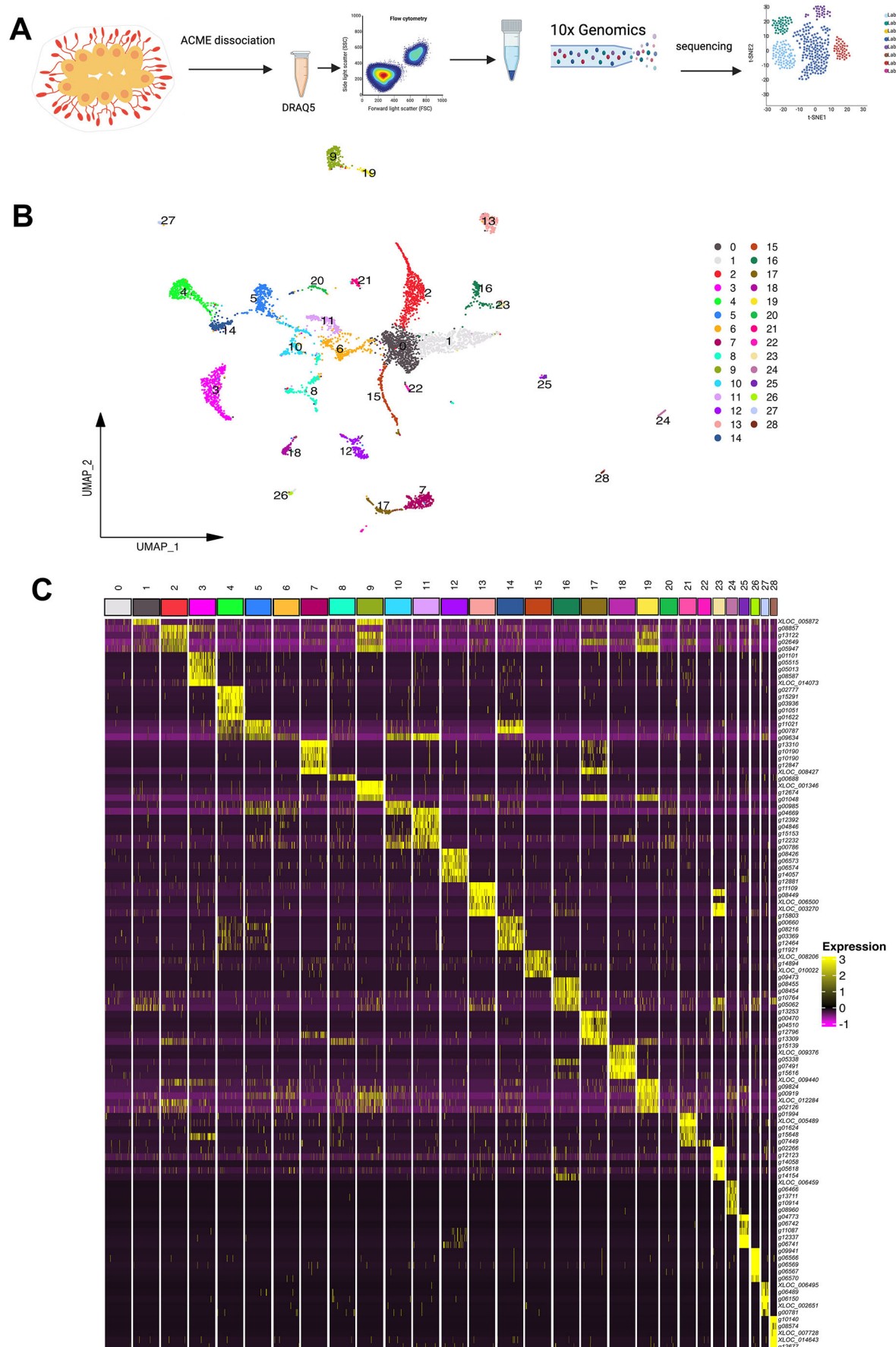

**Fig. 2.** See next page for legend.

**Fig. 2. Overview of the experimental pipeline and clustering results.**
(A) Single cells were prepared using the acetic methanol (ACME) maceration method (García-Castro et al., 2021). Cells were stained with DRAQ5 and sorted by FACS. Cells were captured in droplets using a 10x Chromium system. Single-cell libraries were prepared and sequenced. After mapping the transcripts to the genome, clustering analysis was performed to identify the cell types. (B) UMAP clustering analysis revealed 29 single-cell clusters. Each cluster was color-coded. (C) Heatmap of the top five marker genes in each cluster.

controlling cell shape, movement and transport within the cell (Sewell et al., 2024). Cilia are microtubule-based organelles, and the zone 1 cilium has a different axonemal structure than the other zones (*Ciona* endostyle) (Konno and Inaba, 2020), which may indicate that these cells express distinct combinations of microtubule genes.

GO and pathway analyses were conducted for genes highly expressed in digestive- and branchial-associated tissues (clusters 4, 5, 10, 11 and 14) (Fig. 4). Clusters 10 and 11 were overrepresented in the pathways linked to cilia assembly and movement (Fig. 4). This aligns with the mRNA expression of *Cnfn* and *Tubb*, markers of cell clusters 11 and 10, respectively, which showed intense staining in cilia-rich cell types (Fig. 3C,D). Annotation of clusters 4, 14 and 5 confirmed the enrichment of biological processes associated with food breakdown, such as metabolic and catabolic oxidoreductase activity and cell secretion (Fig. 4).

### Identification of endostyle clusters

Endostyle, a tissue similar to the pharynx, filters food, synthesizes hormones and provides immune defenses (Holley, 1986; Jiang et al., 2023). It is also believed to give a niche to support progenitor cell maintenance (Voskoboynik et al., 2008; Rinkevich et al., 2010; Rosental et al., 2018). Genes known to be expressed in the endostyle of other ascidian species (*Ciona*, *Styela* and *Botryllus*) were selected and identified using the single-cell dataset (Table S4, Figs S5 and S6). Clusters 6, 3 and 8 were deemed potential endostyle clusters based on the expression of known endostyle markers. In the ascidian *Ciona*, galectin is expressed in various regions of the endostyle (Parrinello et al., 2015, 2017). The *B. diegensis* genome contains multiple *Lgal* genes (Table S3) the transcripts of which were detected in clusters 3, 4, 6 and 10 (Table S3, Fig. S7) (Fig. 5A). In *Styela*, *Itnl1/Fcn1* mRNAs were enriched in clusters 8 and 3 and zones 6 and 7 of the endostyle (Jiang et al., 2023). *Muc5a* and *VWF* transcripts were found in clusters 3 and 10 (Sasaki et al., 2003; Yamagishi et al., 2022). Glutathione peroxidase, a common endostyle and branchial sac enzyme (Kobayashi et al., 1983), was observed in our dataset in clusters 3 and 6 (Fig. 5A).

GO and pathway analyses for cluster 3-associated genes revealed enrichment for immunity, ECM, carbohydrate binding and cell adhesion (Fig. 5B). Jiang et al. (2023) also found a list of similar GO terms, including ribosomal, thyroid hormone, immune function, digestive function (such as mucus) and neurosecretory genes (such as semaphorin 1a) for *Styela clava* (Jiang et al., 2023).

The top marker gene for cluster 3, g09775 (Fig. 5C), encodes fibrillin (Fbn1), a protein with fibrillin repeats and an EGF domain that shares 44% identity with the human FBN1, FBN2 and FBN3 proteins. Fbn proteins are found in connective tissues and maintain their elasticity. Fbn1 is crucial for microfibril synthesis, as it binds to calcium and regulates TGFβ release (Reinhardt et al., 1996; Handford, 2000; Chaudhry et al., 2007). *Fbn1* expression was observed in zone 8 of the endostyle (Fig. 5D). This region of the endostyle functions with zone 7 in immune activities, with high iodine and peroxidase activities (Sasaki et al., 2003; Jiang et al., 2023; Alesci et al., 2022). Iodine metabolism has been linked to

thyroid gland evolution (Fujita and Nanba, 1971). We also found that the *Tubb* probe (identified as a ciliated cell marker in Fig. 3D) marked a subset of endostyle cells. *In situ* hybridization showed that its mRNA was present in zone 1 cells of the endostyle (Fig. 5E). These cells have long cilia that, together with mucus, aid in trapping food particles from the water current (Holley, 1986). Based on this information, cluster 3 was assigned as an endostyle cell cluster.

### Identification of blood cell clusters

The top cluster 25 gene *g07537* is predicted to encode a FAD-dependent oxidoreductase domain-containing protein (Foxred2) (Didion et al., 2002). This transcript was also detected in some of the cluster 12 cells (Fig. 6A). Foxred2 balances redox states and contributes to generating reactive oxygen species, which are associated with endoplasmic reticulum stress. Intense *Foxred2* staining was observed in the thin endothelium lining blood vessels and in a small number of immunocytes with a few large granules (Fig. 6B). Additionally, staining was observed in the developing heart tube cells as they became thinner. In ascidians, the expression of Foxred2-like proteins in the vascular lining and granular immunocytes suggests that they may be crucial for maintaining redox balance and metabolic processes that are necessary for vascular function and immune response. This may include protecting cells from oxidative stress, aiding in antimicrobial defense, detoxifying harmful substances and regulating inflammation. GO analysis (Fig. S10) revealed associations with heme binding, biosynthesis, oxidoreductase activity and SLC-mediated transmembrane transport within cluster 25. SLC-mediated transport is important for thin monolayer barriers, such as the blood-brain barrier, to mediate the transport of substances across the endothelium (Morris et al., 2017).

The expression of the cluster 13 marker glutathione S-transferase alpha 3 (*Gsta3*; *g11109*) was also detected in some cells and distributed throughout the other clusters (Fig. 6A). Staining was observed in cytotoxic cells, including granular amoebocytes (Fig. 6C, asterisks), compartment amoebocytes and morula-like cells (Fig. 6C). Morula cells are variable in size, and the extent of staining may have been influenced by vacuole size. Only a few significantly enriched GO terms were identified because of the low number of marker genes in this cluster. However, it included immune system processes (Fig. S10).

The gene g13310 was identified as a cell marker for cluster 17 (Fig. 6A). It is predicted to encode a protein ortholog of CSMD3 with multiple Sushi and von Willebrand factor type A (CUB) domains, typically found in transmembrane receptors or adhesion proteins. Sushi or complement control domains are involved in the immune system (Ermis Akyuz and Bell, 2022). Excessive activation of the complement pathway is prevented by Sushi domain-containing proteins, which bind to activated C3/4 components to target them for degradation (Ojha et al., 2019). *Csmd3*+ cells were detected in the circulation (Fig. 6D), which appeared to be phagocytic cells, including macrophage-like cells and hyaline amoebocytes. Gene ontology analysis revealed an over-representation of terms related to the cytoskeleton, vesicle transport and immunity (Fig. 6F and Fig. S14).

Cluster 8 top marker gene, *g05125* (Fig. 6A), encodes a large protein that shares 30% identity with NOTCH1, NOTCH2 and SNED1 due to multiple Sushi and EGF domains. Several Notch-like genes are present in the *B. diegensis* genome. *In situ* hybridization detected expression in a small number of storage and/or granular cells; larger cells were characterized by multiple small vesicles (Fig. 6E). These cells resemble mast cells and are predicted to release inflammatory factors such as histamine and chemokines (Cima et al., 2001; Blanchoud et al., 2017). GO and pathway

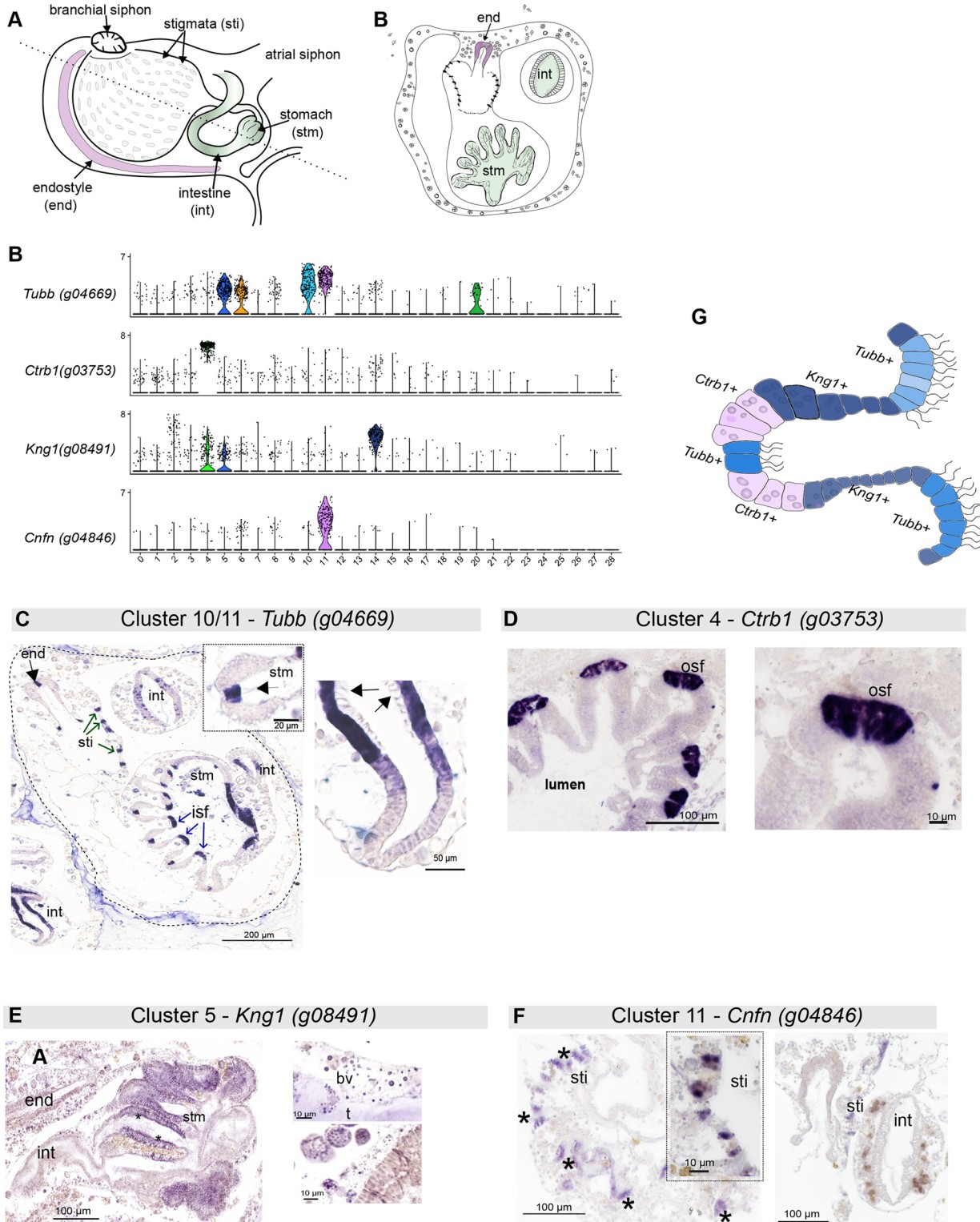

**Fig. 3. Identification of *B. diegensis* digestive tract clusters.** (A) Schematics illustrating an internal view of a single zooid, highlighting the digestive tract and its appearance in a transverse section (right) (dotted line on the left). (B) Stacked violin plot for *Tubb*, *Cnfn*, *Ctrb1* and *Kng1* transcripts, the expression of which was examined by *in situ* hybridization. (C) Cluster expression for *Tubb* identified this gene as a top marker gene for clusters 10 and 11 (B), although it was expressed by cells found in several other clusters. *Tubb* mRNA was found in the inner stomach folds (blue arrows), outer stomach folds (black arrow; inset), stigmata (green arrows) and intestine. (D) *In situ* expression of *Ctrb1* mRNA was observed in the cells on the outer side of each stomach fold. (E) *Kng1*⁺ cells are located in the longitudinal stomach folds and blood cells within the circulation (right panels). (F) Single-cell cluster expression profile for *Cnfn* (*g04846*), the top marker gene for cluster 11 (B). The *Cnfn* probe-stained cells are found in rows (asterisks) that line the branchial chamber, known as stigmata cells. No staining was observed in other tissues, such as the endostyle and intestine (right panel). (G) Based on the *in situ* results, an illustration of a stomach fold showing the location of *Tubb*⁺, *Ctrb1*⁺ and *Kng1*⁺ cells in the stomach epithelium. end, endostyle; stm, stomach; isf, inner stomach fold; osf, outer stomach fold; sti, stigmata; t, tunic; int, intestine; bv, blood vessel; sti, stigmata.

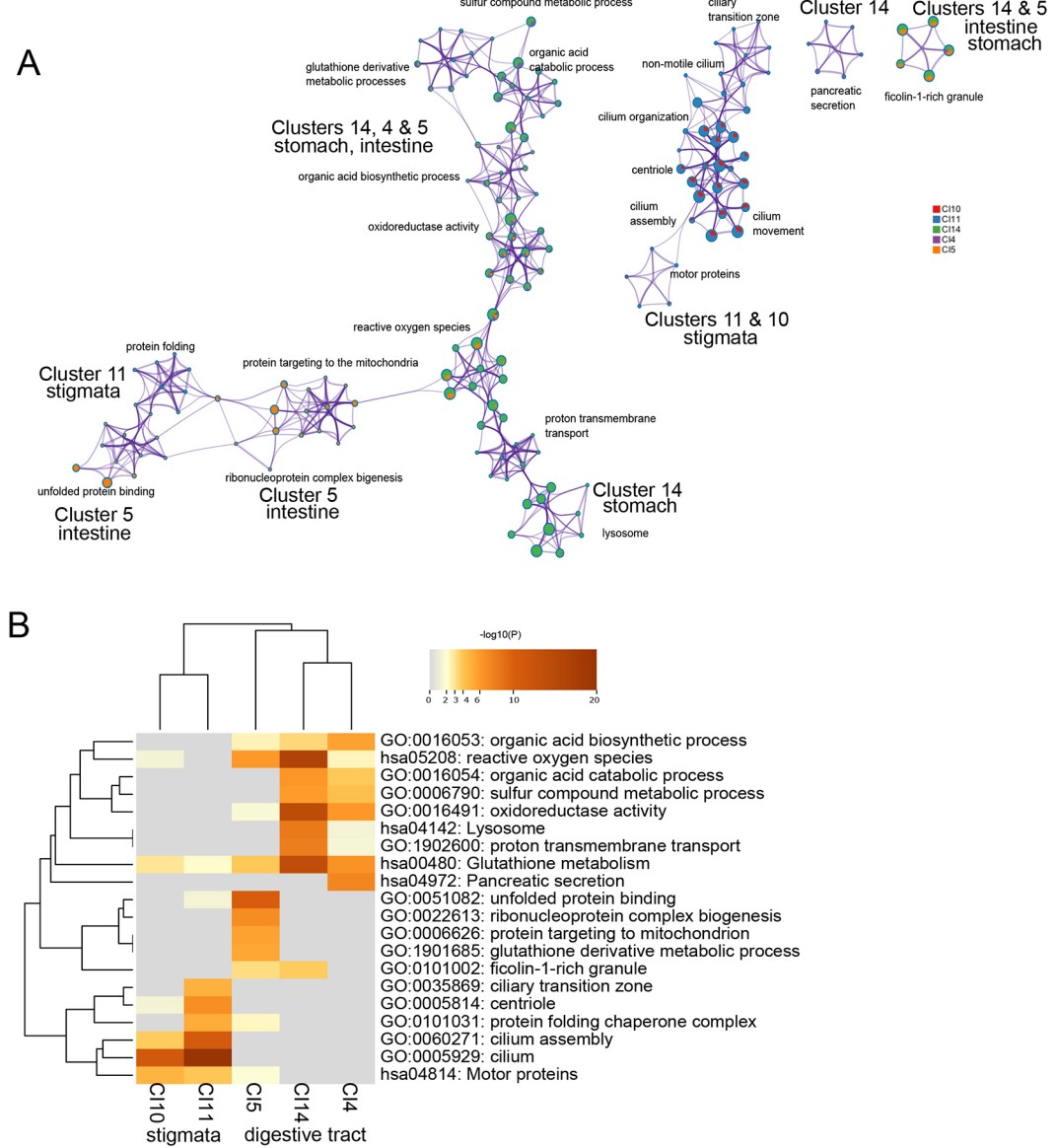

**Fig. 4. GO and pathway analyses for cell clusters associated with digestive functions.** (A) Network plot of enriched biological processes from Metascape analysis. Each node represents a GO or pathway term, and edges indicate functional similarity based on shared gene content or annotation. Nodes are grouped and colored by cluster assignment and predicted tissue type, illustrating how lysosomal degradation, mitochondrial metabolism and ciliary function cluster into broader biological themes across digestive-associated cell types. (B) Heatmap showing −log10(*P*-value) of selected enriched GO terms in clusters associated with the digestive tract and stigmata to highlight distinct biological functions. Each row represents an enriched GO term, and each column represents the gene list for clusters 14, 4, 5, 10 and 11.

analysis indicated terms related to peroxidase and leukocyte-mediated immunity (Fig. 6F and Fig. S16). In addition, we also examined the data for expression of previously identified immunocyte-related genes: *C3* (complement factor), Toll-like receptor genes (TLRs) and C-type lectin genes (CLECs) (Fig. S8). Clusters associated with immune-related genes demonstrated distinct patterns. For example, cluster 8 was enriched for C-type lectin genes, part of a large superfamily implicated in immune recognition (Scur et al., 2023). C3, although not a cluster marker, was expressed in cluster 8, consistent with its role in phagocytic cells. Clusters 9 and 19 exhibited high expression of Toll-like receptors, interferon regulatory factors and other immune-regulatory genes, supporting their involvement in immune functions (Figs S8 and S13). These findings align with the known roles of these markers in tunicate immunity and highlight the complexity of immune-related gene expression in *Botrylloides*.

## Developmental trajectory analysis

To identify the cells forming the bud disc of the peribranchial epithelium, we focused on cells with the highest expression of transcription factors found in disc cells: *Pitx1*, *Otx*, *Nk4* and *Runx* (Tiozzo et al., 2005; Langenbacher et al., 2015; Ricci et al., 2016). The subset of cells within cluster 6, likely the peribranchial cluster, showed the highest joint density, indicating that these cells co-expressed these four genes (Fig. 7C). Using this as the root, the predicted cell trajectory was plotted using Monocle3. Pseudotime trajectory paths indicate the progression and pathways of these cells, going from this putative stem cell cluster to more differentiated states later in the pseudotime. This analysis connected all Seurat clusters, with the origin located in cluster 6 (Fig. 7C). To extend this further, we determined whether several previously studied candidate stem cell markers were present in the scRNA-seq dataset, including *Itga6*,

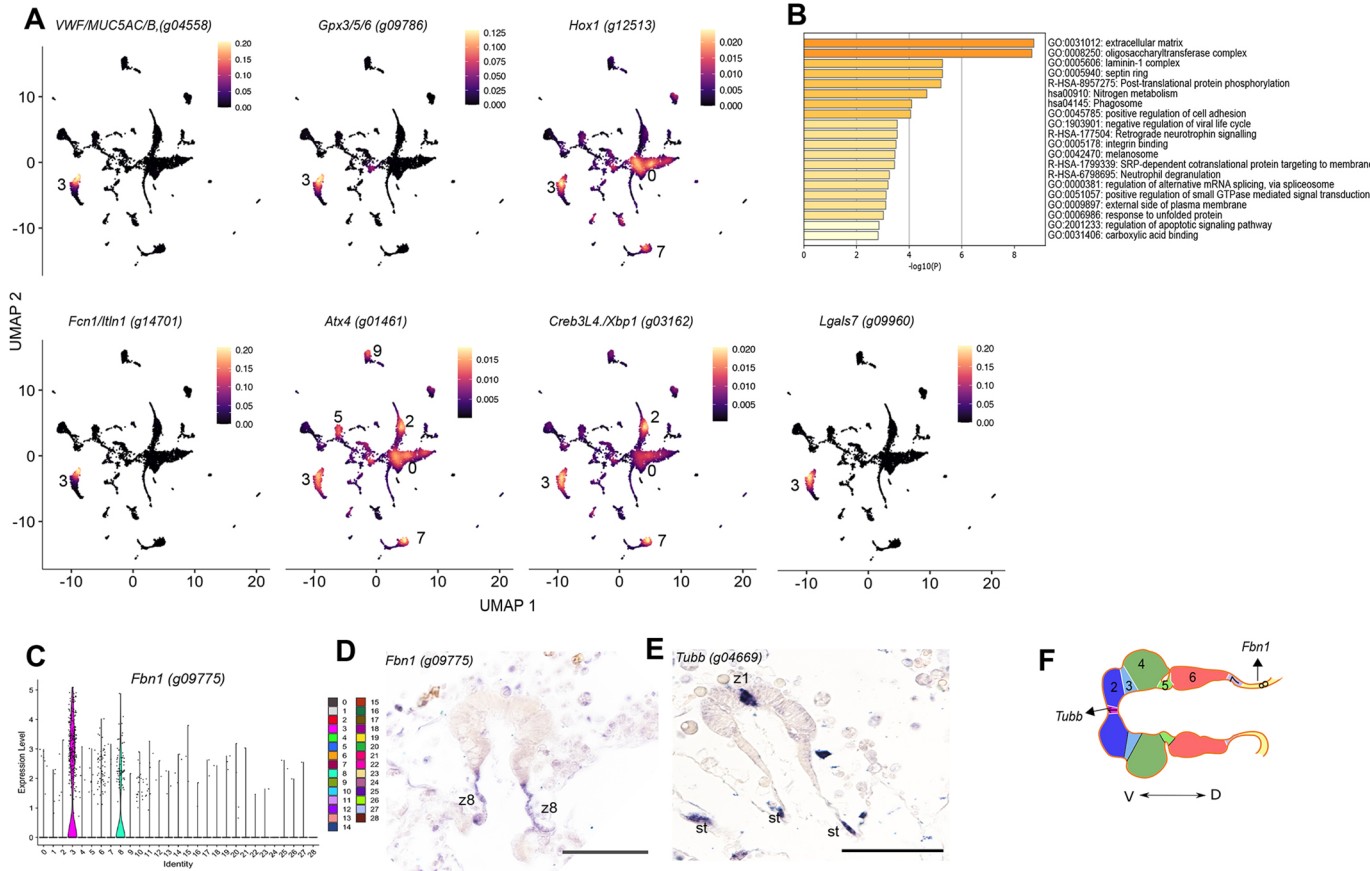

**Fig. 5. Identification of endostyle clusters.** (A) UMAP feature density plots show the highest concentration of cells expressing each of these tunicate endostyle genes. (B) GO and Pathway terms colored by *P*-value, as determined by Metascape. (C) Violin plot for cluster marker 3, *Fbn1*. (D) *In situ* expression pattern for the cluster 3 marker *Fbn1* in zone 8 of the endostyle. (E) *Tubb* mRNA was detected in zone 1 cells of the endostyle. (F) Schematic showing the different zones of the endostyle, highlighting the spatial expression of *Fbn1* and *Tubb*. z8, zone 8; z1, zone 1; st, stigmata; V, ventral; D, dorsal. Scale bars: 50 µm.

*Notch2*, *Vasa* and *Piwi* (Kawamura and Sunanaga, 2010; Rinkevich et al., 2010; Rosental et al., 2018; Kassmer et al., 2020). *Vasa*, *Piwi1* and *Piwi2* were missing from the dataset. *Notch2* and *Itga6* were detected in the cells scattered across several clusters (Fig. S17). Additionally, orthologs of the reprogramming factors (Yamanaka factors) (Takahashi and Yamanaka, 2006, 2016), *Oct3* and *Oct4* (POU3 and POU4 proteins), *Sox2* (SOXB subgroup of Sox factors), *Klf4* and *Myc* were found in the same group of cells within cluster 6 (Fig. S18). This further supports the notion that a subset of cluster 6 cells likely has stem cell properties and represents the bud disc.

The trajectory of the colony dataset was analyzed to understand the relationship between cell clusters and differentiation time using Monocle3 (Fig. 7D). The trajectory of cell development was charted on UMAP, with the candidate progenitor bud disc cells of the peribranchial cluster (cluster 6) serving as the root (Fig. 7C). Pseudotime trajectories reveal that the clusters, predicted to be bud, peribranchial and brachial epithelial cells, emerge early. In contrast, clusters associated with specialized tissues, such as the stomach epithelium, appear later in the pseudotime (Fig. 7D). Epithelial tissues within the zooid appeared early in the trajectory, preceding the development of stomach and endostyle tissues (left branch). The third pathway leads to immune cell formation. Overall, the development of zooid-related tissues later in the trajectory is consistent with blastogenesis (Fig. 7A,B).

To further investigate cluster 6, we cloned probes for two top-ranked markers, *Col24a1* (*g02151*) and *Igal4/7* (*g08355*). Both transcripts show the highest expression in cluster 6, with scattered positive cells in other clusters (Fig. 8A-C). *Col24a1* is strongly expressed in the peribranchial epithelium and blastodiscs at early developmental stages but is excluded from the thicker epithelium undergoing organogenesis (Fig. 8Di-iii). Expression continues in the epithelium, blastodiscs and forming stigmata, and is notably strong in the outer vesicle epithelium, which ultimately forms the zooid epidermis (Fig. 8Div-v). *Col24a1* is absent from fully developed zooid tissues, such as the stomach epithelium (Fig. 8Dvi,vii). Within the vessel epithelium, *Col24a1* is strongly expressed in the tunic chamber-associated VE, and more weakly in the connecting vessel (to the zooid) and the perivisceral epithelium lining the gut (Fig. 8Dvi, orange arrowheads). Mature blood cells lack *Col24a1* expression (Fig. 8Dviii-x, asterisks), compared to smaller epithelial-associated cells (Fig. 8Dix,x), possibly reflecting maturing blood cells that lose *Col24a1* upon terminal differentiation.

*igal4/7* displays a broadly similar pattern of expression (Fig. 8E). Little expression in early buds and stronger staining was observed, particularly in the outer vesicle epithelium, associated blood cells and the PBC epithelium (Fig. 8Ei-iii). In primary buds, *igal4/7* is strongly expressed in the folded branchial epithelium stigmata precursors (Fig. 8Eiv). In adults, only weak expression remains in some regions of the gut epithelium (Fig. 8Ev, boxed region) and zone 8 of the endostyle (Fig. 8Evii). Blood cells bordering the vascular epithelium (Fig. 8Evi-viii, orange arrows) appear stem-like, with a large nucleus to cytoplasm ratio (Fig. 8Evi, inset).

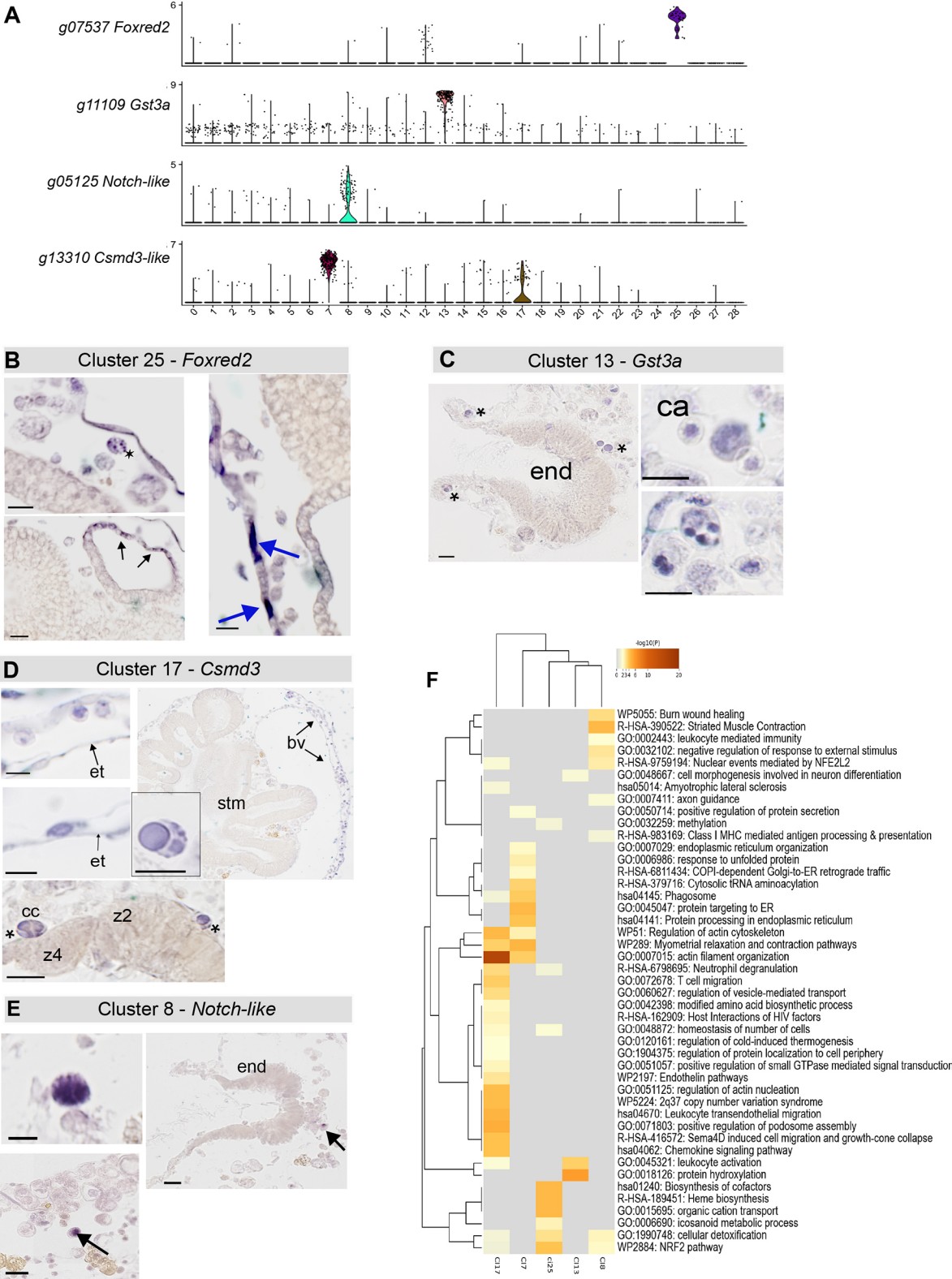

**Fig. 6. Candidate blood cell clusters.** (A) A stacked violin plot representing the Seurat cluster expression of four marker genes, further examined by *in situ* hybridization. (B) Detection of *Foxred2* mRNA by *in situ* hybridization identified vascular endothelium (blue arrows), and cells of the developing heart and some blood cells (asterisk). (C) Cluster 13 g11109 *Gst3a* marker *in situ* hybridization. Positive cells were found in the circulatory system and near the endostyle (asterisks). (D) Cluster 17 marker, *Csmd3*, mRNA-positive cells were in the vasculature, near the endostyle (asterisks) and the thin epithelium lining the tunic vessels (arrows). (E) Cluster 8 marker, *g05125* (*Notch-like*), probes stained only a few scattered cells in the vascular circulation stained with the *g05125* probe (arrows). (F) GO and Pathway heatmap showing enriched biological processes in clusters 17, 7, 25, 8 and 13. A heatmap is shown to enable direct comparison across blood-associated clusters, highlighting shared versus distinct functional associations. stm, stomach; et, vessel endothelium; bv, blood vessel; end, endostyle; ca, compartment amoebocyte; cc, compartment cell; z2, zone 2; z4, zone 4. Scale bars: 10 µm.

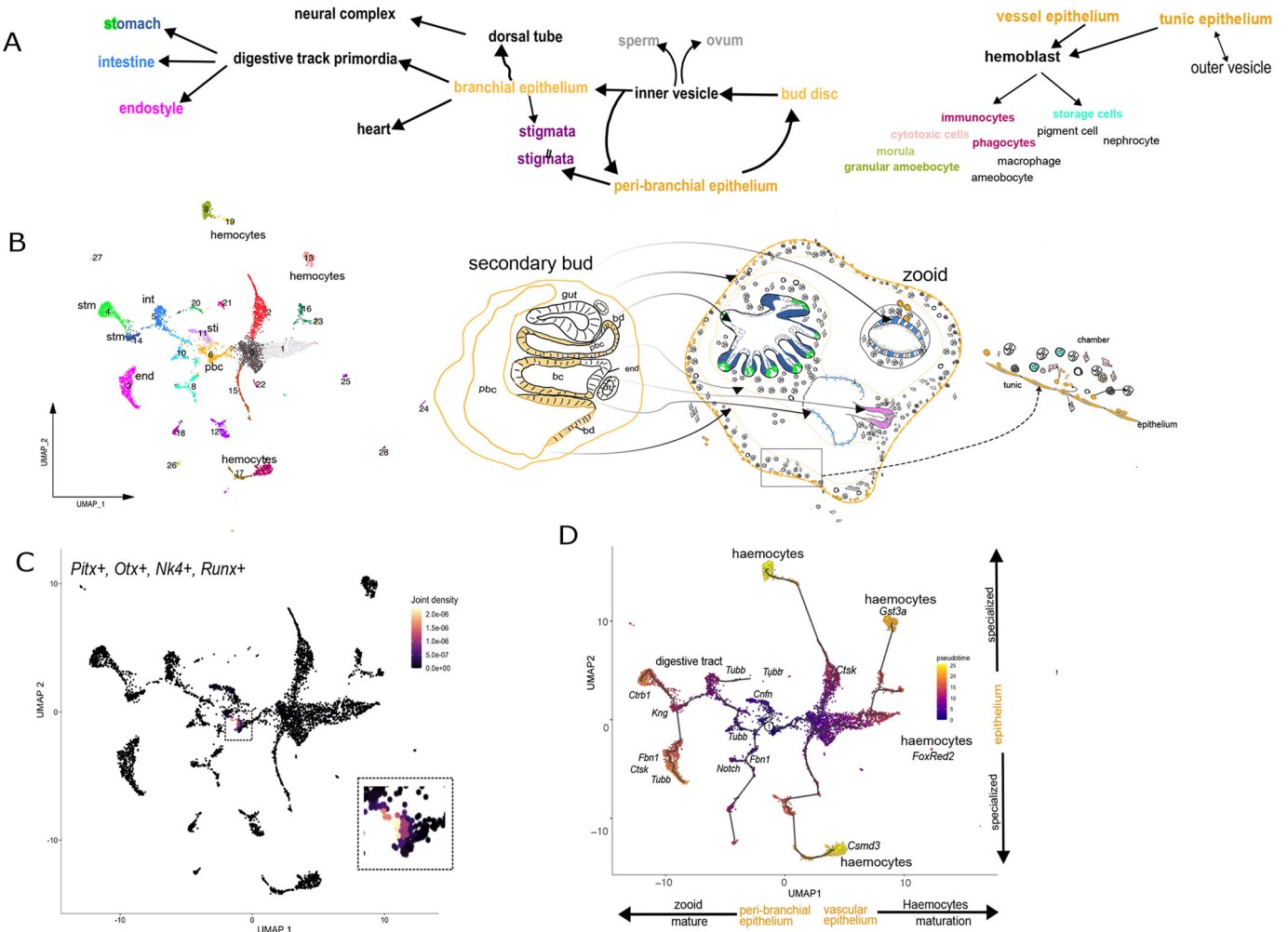

**Fig. 7. Cell populations and their relationships.** (A) Schematic overview of the principal tissues and organs in a *B. diegensis* colony and their proposed origins. Budding arises from a small group of cells in the peribranchial chamber epithelium of the immature zooid (before stigmata perforation), which forms the bud disc (Berrill, 1947). This disc separates from the parent epithelium to create a closed vesicle encased by the outer epidermis of the parental zooid. Germ cells segregate early from the inner vesicle to form oocytes and sperm. On the dorsal side of the vesicle, an invagination produces a neural placode. The heart and digestive tracts form on the posterior side, with organ primordia first appearing as outpockets of the branchial epithelium. Cells lining blood vessels, ampullae and epithelia have also been proposed to serve as hemocyte sources (Rosental et al., 2018; Rinkevich et al., 2010). (B) Representative schematic showing relationships among these tissues, placed beside a UMAP cluster plot. The arrows indicate transitory relationships between various epithelial compartments. (C) UMAP density plot illustrating the co-expression of *Pitx*, *Otx*, *Nk4* and *Runx* within cluster 6. Cells with higher levels of co-expression are rendered in increasingly orange tones. (D) Monocle 3 trajectory analysis. The trajectory root is inferred to be the progenitor population (cluster 6), most likely corresponding to the peribranchial epithelium. end, endostyle; sti, stigmata; pbc, peribranchial chamber; bc, branchial chamber; int, intestine; stm, stomach; bd, bud disc; dt, dorsal tube.

For comparison, we examined two terminal differentiation markers, *Tubb* and *Ctrb1* (Fig. 8F,H), at early stages of development. Both show no expression in bud or secondary bud tissues (Fig. 8Fi,ii and Fig. 8Hi,ii). *Tubb* is upregulated in the primary bud stomach cells begin to specialize, first in the inner fold cells (Fig. 8Fiii). *Ctrb1* transcripts are detected in the mature stomach folds. Collectively, these observations indicate that cluster 6 harbors progenitor cells, including peri-branchial epithelium and vascular epithelium, with *Col24a1* and *igal4/7* expression diminishing as differentiation proceeds.

To explore the relationships between clusters, we generated PAGA plots to investigate the underlying structure and infer potential differentiation pathways (Fig. 9A). Each node in the plot represents a distinct cell cluster, and lines connecting the nodes indicate lineage relationships. We observed that certain clusters (e.g. 20, 21, 0 and 6) occupy central nodes with multiple outgoing connections, reflecting high transcriptional connectivity to numerous other clusters. In lineage-inference approaches, this pattern commonly signifies progenitor-like states that branch toward multiple differentiated fates (Trapnell et al., 2014; Wolf et al., 2019; Bergen et al., 2020).

In addition, a dpt pseudotime visualization (Fig. 9B) demonstrates how cells are distributed along a developmental timeline, with darker colors indicating earlier states and lighter colors representing more advanced states. This supports the conclusion that cells in the highly connected clusters (20, 21, 0 and 6) occupy earlier positions in pseudotime, consistent with progenitor properties.

The velocity (scVelo) analysis uses spliced-to-unspliced mRNA ratios to illustrate the speed and direction of cell transitions. Longer arrows indicate faster transcriptional shifts, whereas shorter arrows suggest more stable states. Convergence or divergence of arrows at specific points provides insight into lineage decisions. Notably,

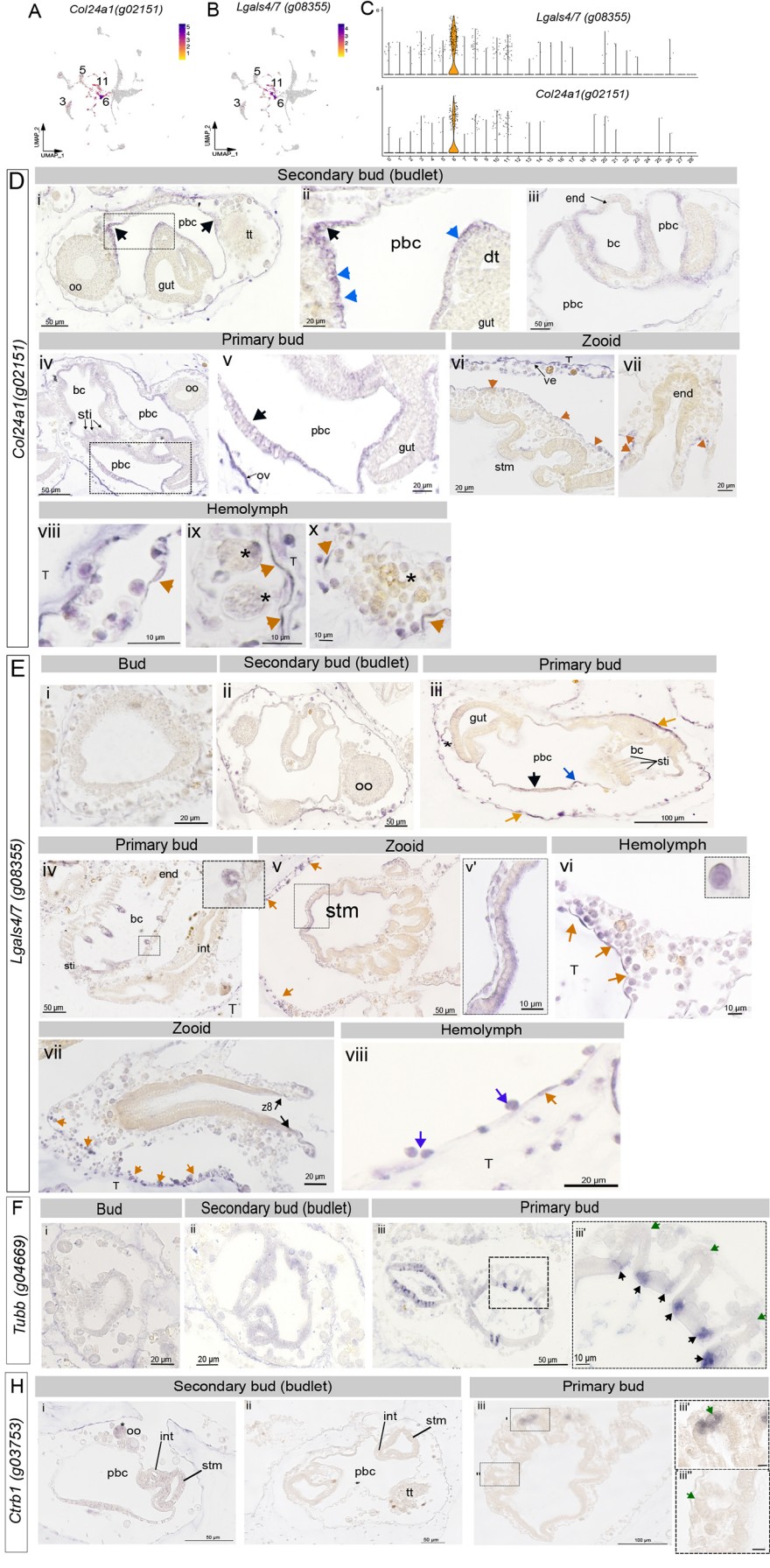

**Fig. 8. Expression of *Col24a1* (*g02151*) and *Lgal4/7* (*g08355*) during colony development in *B. diegensis*.** (A,B) UMAP plots showing *Col24a1* and *Lgal4/7* expression across cell clusters. (C) Violin plots highlighting the significant enrichment of these genes in cluster 6. (D) *In situ* hybridization for *Col24a1*. (i,ii) In the secondary bud ('budlet'), a strong signal is present in the PBC (pbc)epithelium, including the blastodisc (black arrows). tt, testis; oo, ovary; dt, dorsal tube. (iii) As the branchial chamber (bc) epithelium thins, *Col24a1* persists there but is absent from the endostyle (end). In the primary bud (iv,v), expression continues in the PBC and BC epithelium, and in the outer vesicle (ov) epithelium and blastodisc. sti, stigmata. In the zooid stage (vi, vii), no staining is observed in gut tissues; however, strong mRNA signal appears in cells associated with the vessel epithelium (ve) on the tunic-vessel side (contrasting the vessel-chamber side) and in the perivisceral epithelium (orange arrowheads in vi, vii). stm, stomach. (viii-x) In the hemolymph, *Col24a1* localizes to smaller cells that resemble hemoblasts/ stem-like cells or immature immunocytes, as well as thin vessel epithelium-lining cells (arrowheads). Mature immunocytes (morula cells; asterisks in ix, x) show no expression. T, tunic. (E) *In situ* hybridization with *Lgal4/7* RNA probe. (i) Only faint staining is detected in the early bud. (ii,iii) Stronger expression emerges in the thinning PBC epithelium (arrows) and the outer vesicle layer (orange arrows). (iv-vii) In the primary bud (iv), *Lgal4/7* marks the stigmata primordia in the BC epithelium; it is also present in the maturing stomach epithelium (v) and appears faintly in zone 8 of the endostyle (vii). Blood cells adjacent to the VE are also strongly positive (orange arrows in v and vi). (viii) Expression is seen in cells attached to the VE (blue arrows), with some detaching into the vessel lumen and others localizing in the tunic (orange arrow). (F) Expression of the marker gene *Tubb* (g04669) during blastogenesis. (i,ii) No expression is observed in the initial or secondary buds. (iii,iii′) *Tubb* appears first in the developing gut and intestine – initially in the inner stomach folds (black arrows) and then in the ciliated cells at the top of the outer stomach folds (green arrows; see also Fig. 3). (H) Expression of the cluster 4 marker *Ctrb1* (*g03753*) in secondary and primary buds. (i,ii) No signal is detected in the early, pre-folded stomach. (iii,iii′) Once the stomach loops fold, *Ctrb1* is expressed in specialized cell types of the maturing stomach (green arrow), while newly folded loops still lack this staining (green arrow, iii″). z8, zone 8. Scale bars: 10 µm in Dviii-x, Ev′, Fiii′, Hiii′,iii″; 20 µm in Dvi,vii, Ei,vii,viii, Fi,ii; 50 µm in Eiii, iv,v, Fiii, Hi,ii; 100 µm in Eiii, Hiii.

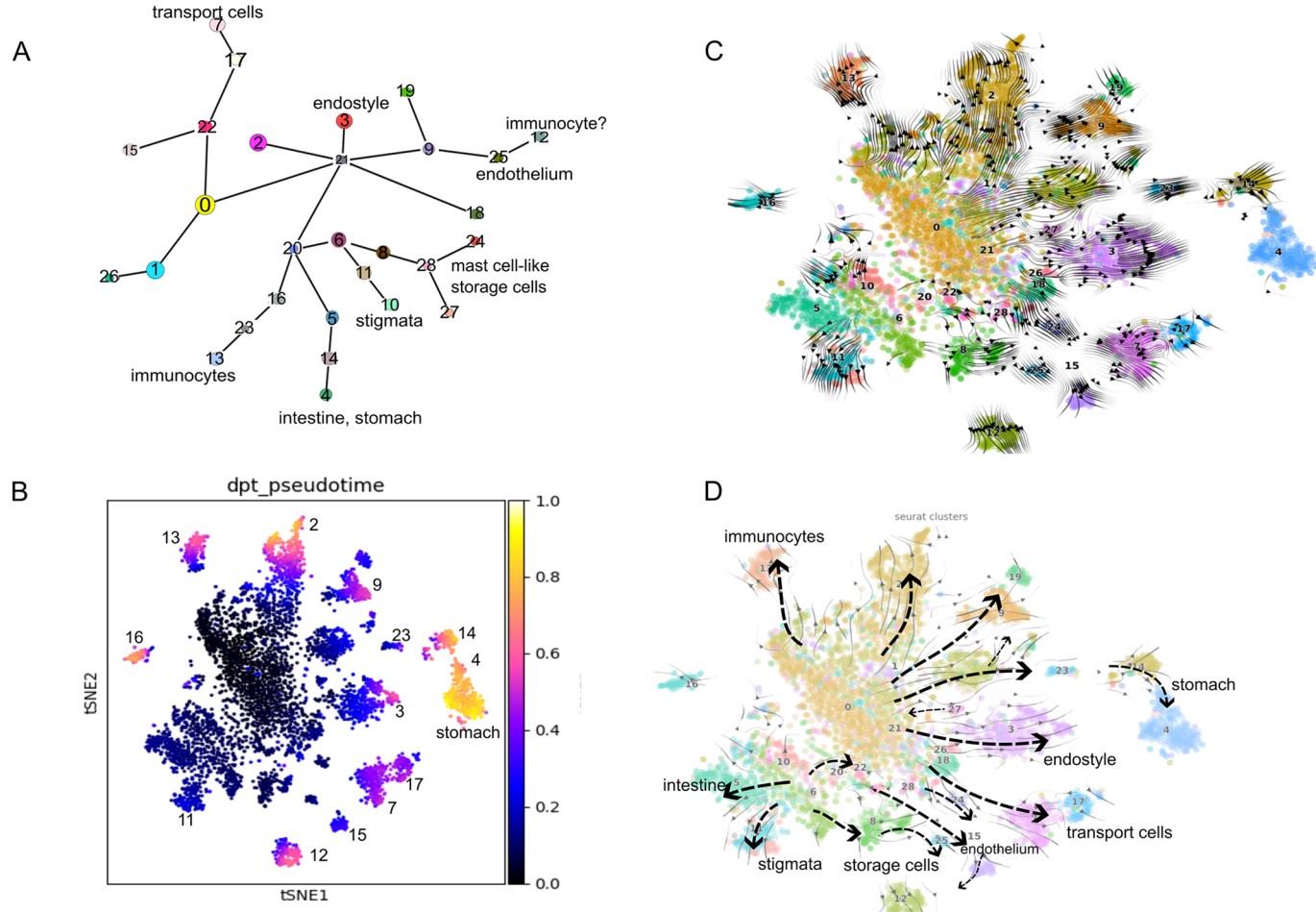

**Fig. 9. Identification of potential differentiation pathways using CellRank.** (A) The PAGA plot displays each cell cluster as a node, with branching lines indicating potential differentiation pathways among the clusters. (B) A tSNE projection where cells are colored by pseudotime, showing the progression of cell states over time. Darker colors represent earlier pseudotime states, whereas lighter colors represent later pseudotime states. The Cluster IDs are given for the later states. A UMAP projection is shown in Fig. S20. (C,D) Velocity plots using the tSNE projection, where arrows represent the direction of RNA velocity, indicating the predicted future states of the cells based on their transcriptional activity and pseudotime. The cells are colored using the Seurat Cluster ID. (D) An annotated version of plot C shows a detailed identification of specific cell types and their predicted differentiation pathways.

clusters with numerous outgoing arrows (e.g. 0, 6, 21 and 20) function as central nodes for branching (Fig. 9C), mirroring their positions in the PAGA graph (Fig. 9A) and underscoring their likely progenitor roles.

Finally, CellRank was used to examine cell fate dynamics (Fig. 10 and Figs S19, S20). It has been designed to handle complex data, such as that produced from developing and regenerating tissues, which involve many initial, intermediate and terminal cell states (Lange et al., 2022). Fig. 10C and Fig. S14 present bar graphs showing the aggregate fate probabilities of cells from the Seurat cluster. These cells will progress to a terminal state, while the clusters with multiple bars of varying heights have different probabilities of transitioning to different end states, such as clusters 0, 5, 6, 8, 20, 18, 21, 22, 26 and 28. In contrast, clusters with cells that primarily transition to one fate display a single high bar. For example, all cells in cluster 11 transitioned to the ciliated stigmata cell type, according to GO and *in situ* expression (Fig. 3C).

We used CellRank to identify potential driver genes strongly associated with the terminal state that may serve as key regulators or markers of cell fate (Fig. 10E and Table S4). These genes exhibited high expression levels at the earliest pseudotime but were low in the final terminal state. Such genes may play a role in initiating or driving

the early stages of differentiation and decreasing their expression as cells reach their final differentiated state. One example of a potential driver gene is the *Lin28* ortholog, a highly conserved RNA-binding protein that regulates stem cell differentiation and proliferation in organisms ranging from nematodes to mammals (Wu et al., 2022). This gene is expressed in undifferentiated pluripotent cells and is downregulated as cells progress toward a specialized cell fate or undergo reprogramming (Pieknell et al., 2022). Another potential driver gene is the *Atf4* and/or *Atf5* ortholog. In vertebrates, this transcription factor is associated with stress-induced responses and the differentiation of stem cells into secretory cells. It plays a role in epithelial differentiation and stresses immune response (Barrera-Lopez et al., 2024). *B. diegensis g11626* encodes a DExD/H-box polypeptide: 39 B (DDx39b)-like RNA helicase. DDx39b regulates mRNA splicing events, including the processing of transcripts of genes required for immune and myocyte cell fate (Zhang et al., 2018; Hirano et al., 2023). Another potential driver gene is high mobility group box 1 (*Hmgb1*), a chromatin-associated protein secreted by immune cells and involved in differentiating mesenchymal stem cells into vascular cells (Meng et al., 2018).

We applied the StemID2 pipeline to the Seurat dataset, using the Seurat clusters to enhance our understanding of the lineage

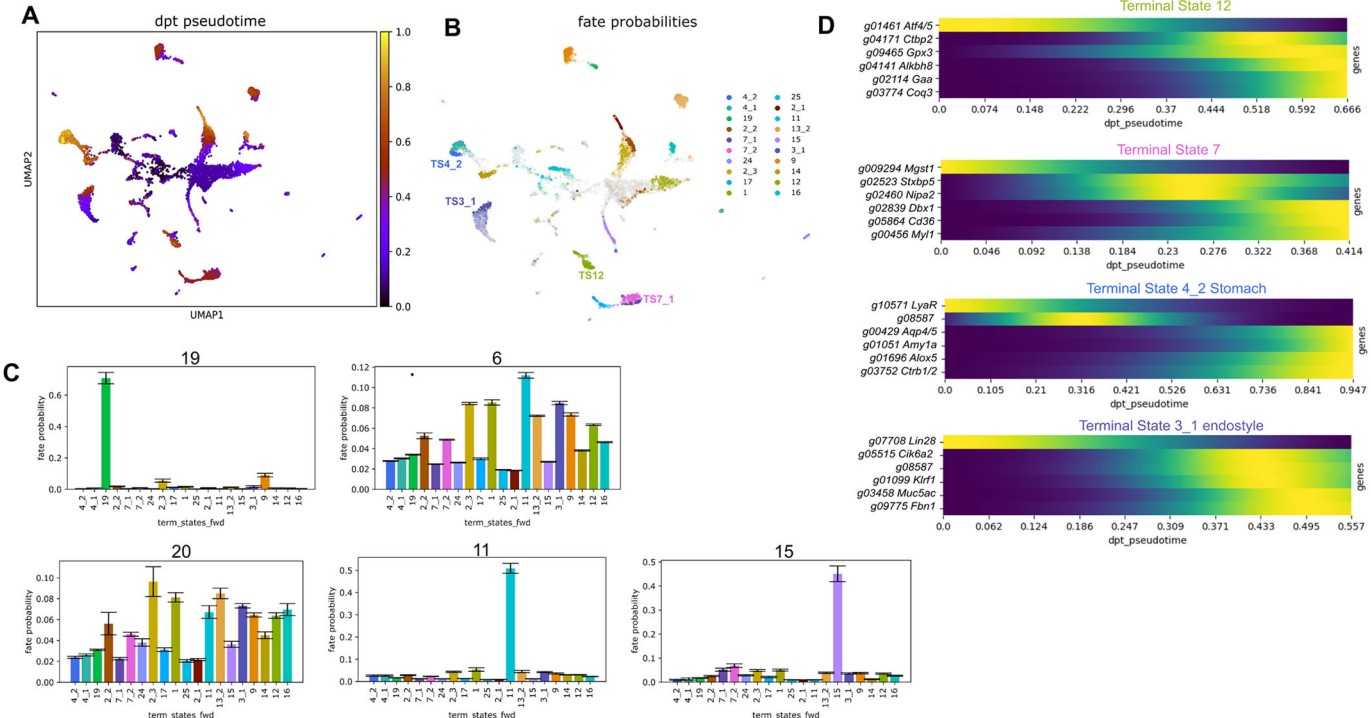

**Fig. 10. Cell fate dynamics.** (A) UMAP plot showing diffusion pseudotime (dpt_pseudotime). The color scale represents pseudotime, with cells colored yellow appearing later in the predicted cell trajectory pathway. (B) Fate probability UMAP, shown as terminal clusters. TS, terminal state. (C) Aggregate fate probability bar graphs. These display the probabilities of different clusters (20, 11, 15, 19 and 6) progressing toward the terminal states, as predicted by CellRank. (D) A selection of driver genes is presented alongside a gene whose expression is highest at the earliest pseudotime on the lineage towards the terminal state fate (Table S4).

relationships among clusters and potential progenitor populations. Due to internal reindexing by RaceID (which starts cluster numbering at 1), Seurat cluster 6 is referred to as cluster 7 in the StemID output.

The lineage graph revealed that cluster 7 (Seurat cluster 6) occupies a central position and establishes multiple connections to clusters 1, 2, 6, 9, 11 and 12 (Seurat clusters 0, 1, 5, 8, 10 and 11) (Fig. S21A). This cluster had the highest StemID score, consistent with a progenitor population (Fig. S21B). Clusters 1 and 2 (Seurat clusters 0 and 1) were positioned directly downstream, displaying intermediate StemID scores, suggesting they represent transitional states (Fig. S21B).

Together, these analyses provided strong supporting evidence for the inferred differentiation pathways. They identified key transitional cell clusters and several candidate progenitor cell populations.

## DISCUSSION
Using scRNA-seq, we characterized the cellular composition of an entire *B. diegensis* colony, identifying 29 clusters spanning zooids, buds, blood cells and vascular structures. As an extant colonial chordate, *B. diegensis* holds a unique phylogenetic position near vertebrates, making it a valuable comparative model. This sc-RNAseq resource paves the way for examining the cellular composition of the colonial chordate *B. diegensis*.

### Insights from Gene Ontology and Pathway analysis
GO and Pathway analyses of the top marker genes for each cluster provided a helpful starting point for functional annotation, revealing conserved gene regulatory programs in *B. diegensis*. For example, clusters 9 and 19 showed significant enrichment for terms related to phagocytic vesicles, lysosomes and immunological synapses – consistent with a hemocyte-like lineage. Similarly, clusters 10 and

11 were enriched for ciliary processes, corroborating our *in situ* hybridization results indicating that *Tubb* and *Cnfn* are expressed in highly ciliated epithelia. Despite these insights, GO analysis has limitations. First, many annotations rely on the 'closest' vertebrate or human ortholog, an approach that may overlook species-specific roles or misinterpret homology in tunicates. Second, if a cluster primarily comprises transient or intermediate cell states, its marker genes may not fall into neat functional categories. As a result, certain clusters – particularly those with broadly expressed transcripts – did not yield strong enrichments for any one function. These patterns echo previous studies in other colonial ascidians that describe similarly 'ambiguous' blood cell types or unannotated genes with limited homology to model organisms (Rosental et al., 2018). Thus, while GO analysis can pinpoint broad functional themes, it must be interpreted cautiously, especially in organisms where many genes remain uncharacterized, and dynamic and transitory cell populations are likely abundant.

### Progenitor-like epithelial populations
Adult stem cells are involved in various epithelial tissues in invertebrates. For example, in sponges, epithelial cells function similarly to stem cells and play a role in tissue repair (Rinkevich et al., 2022). This observation suggests a conserved mechanism across different invertebrate species, where epithelial cells have multipotent capabilities that are essential for regeneration and repair. This comparison underscores the importance of epithelial and vascular progenitors in colonial ascidians and other invertebrates, indicating a potentially conserved evolutionary strategy for tissue regeneration and maintenance. We hypothesized that epithelial- and vascular-associated progenitors play key roles in the development and maintenance of colonial ascidians.

Trajectory analysis of cell fate during blastogenesis suggests that multiple initial and intermediate states are present within the colony at any given time. In many ascidians, bud development is initiated by the emergence of a bud primordium through the thickening of the peribranchial epithelium, which is proposed to house multipotent epithelial cells (Tiozzo et al., 2005; Kawamura et al., 2008; Kurn et al., 2011). Transdifferentiation and dedifferentiation have not been formally demonstrated in colonial ascidians such as *Botrylloides* or *Botryllus*, but related processes have been described in other budding species. For example, in *Polyandrocarpa misakiensis*, evidence supports the transdifferentiation of peribranchial epithelium into new bud tissues (Fujiwara and Kawamura, 1992; Kawamura et al., 2008; Shibuya et al., 2015). Based on our scRNA-seq data, we hypothesize that cells from the peribranchial epithelium may be represented in cluster 6 and that these contribute to early bud development, potentially via direct reprogramming or the maintenance of a multipotent epithelial state.

Further evidence that cluster 6 represents an early, progenitor-like epithelial population comes from the expression patterns of two top markers, *Col24a1* and *Lgal4/7*. *In situ* hybridization shows that both genes are strongly expressed in the peribranchial and branchial epithelia during bud initiation and early bud stages, yet largely absent from fully differentiated tissues such as the mature gut. Notably, *Col24a1* and *Lgal4/7* also appear in small cells associated with the outer vesicle epithelium, stigmata precursors, and the developing vascular network – areas where progenitor-like activity is expected. In keeping with the trajectory analysis (which places cluster 6 at the outset of a differentiation continuum), the restricted expression of *Col24a1* and *Lgal4/7* in early bud structures supports the idea that cluster 6 contains multipotent epithelial cells that contribute to multiple adult tissues. This finding aligns with the notion that the colony epithelium serves as a dynamic reservoir of progenitor cells, driving both the formation of new buds and the ongoing turnover of zooid tissues.

The endostyle has been proposed as a stem cell niche in zooids, functioning as a transitional site where stem cells home in and differentiate into blood cells that support the zooid. This niche is considered transitional due to the 2-week cycle of zooid turnover, during which mature zooids are replaced by new ones, requiring continuous replenishment of blood cells to sustain colony function (Voskoboynik et al., 2008). Cells associated with the blood vessel lining, ampullae and epithelial tissues have been proposed as stem cell sources (Rinkevich et al., 2010; Rosental et al., 2018; Vanni et al., 2023 preprint). These findings indicate that ascidian stem cells are not fixed in one location but instead transition between different niches within the organism. The transitional nature of stem cell niches, such as the endostyle, underscores the dynamic nature of stem cell populations in colonial ascidians.

### Study limitations

Assigning cellular identities in scRNA-seq is inherently challenging due to the complexity and variability of gene expression profiles. The study relied on a limited set of ~6000 cells, which may not capture the full diversity of cell types in the organism. Low-abundance transcripts might be under-represented, leading to potential bias in gene expression data and overlooking genes expressed at low levels. Obtaining cells from the tunic, which is rich in cellulose, poses a significant challenge, resulting in the absence of certain tunic cell types. The timing of colony collection may have missed some stages of the sexual cycle, potentially omitting important cell types or developmental stages. For example, stages of oogenesis will be missed as *B. diegensis* only

sexually reproduces briefly at the end of the summer in New Zealand.

Our analysis of immune-related markers highlights both the potential and the limitations of assigning cluster identities in *Botrylloides*. While markers such as C3 and TLRs offer insights into the roles of phagocytic cells and immune regulation, their variable expression across ascidian species complicates direct interpretation. For example, C3 is broadly expressed in immune cells but is not a definitive marker for a single cell type. Similarly, the large number of C-type lectin genes in the genome suggests a diversity of immune functions that warrant further exploration. Integrating consistent functional evidence with expression data will establish a robust framework for annotating immune-related clusters in *Botrylloides*.

*In situ* hybridization, which is useful for the spatial localization of gene expression, is challenging to quantify accurately, limiting its utility in confirming single-cell RNA sequencing (scRNA-seq) findings. No one cluster was identified as marked by gene expression commonly attributed to stem-like cells in colonial ascidians (Ballarin and Rosner, 2022). Specifically, previously identified germline markers for colonial and solitary tunicates, such as *Piwi* or *Vasa* (Rinkevich et al., 2010; Kawamura et al., 2011), were missing from the dataset. This absence could be due to technical dropout, a common issue in scRNA-seq where lowly expressed genes might not be detected in every cell, leading to their apparent absence in the data (Kharchenko et al., 2014). The genes in *B. diegensis* were named and assigned functions for gene ontology analysis based on their closest vertebrate orthologs, assuming similar functions in ascidians. However, many genes lack orthologs, complicating functional prediction.

### Summary

We present a single-cell transcriptomic atlas of a mature *B. diegensis* colony, revealing 29 major cellular states encompassing key zooid, bud and vascular compartments. The data highlight early progenitor-like cells in peribranchial epithelia (cluster 6), multiple blood cell lineages, and specialized digestive and endostyle clusters. These findings open new avenues for exploring tissue function, regeneration and the evolutionary position of colonial ascidians among chordates. Our results underscore the power of scRNA-seq in dissecting the cellular architecture of complex colonial organisms.

## MATERIALS AND METHODS
### Animal husbandry

Colonies of *B. diegensis* were collected from Otago Harbor in New Zealand (45°52′18.1″S, 170°31′37.6″E) and attached to 5×7 cm glass slides. The tanks were aerated and the colonies were fed a shellfish diet (a blend of marine microalgae) with regular seawater changes. Colonies were confirmed to be *B. diegensis* by COI barcoding (Temiz et al., 2023).

### Single-cell preparation and fluorescence-activated cell sorting

Single cells of *B. diegensis* were prepared using the acetic methanol (ACME) dissociation method described by García-Castro et al. (2021). Fresh ACME solution was prepared with DNase/RNase-free distilled water, methanol, glacial acetic acid and glycerol at a ratio of 13:3:2:2. The mature colony was in stage A of blastogenesis with active filtering, newly emerged primary buds and no signs of secondary buds. The colony was placed in a Petri dish on a glass slide, where it was attached using a microtome razor blade with minimal disturbance. The animal was washed with 1 ml of 7.5% N-acetyl l-cysteine in 1×PBS. This solution removes excess seawater and protects RNA. ACME (1 ml) was added, and the colony was minced well using a single-edged razor blade.

The suspension and all the larger tissue pieces were placed in a 1.5 ml tube. The tube was placed in a rotator to apply seesaw motion at ~30-40 rpm for 1 h at room temperature. The sample was pipetted twice and strained

with a 40 μm cell strainer into an ice-covered 50 ml Falcon tube. The cells were cooled to prevent RNA degradation. The suspension was centrifuged at 1000 *g* for 5 min at 4°C. The supernatant was removed, and the pellet was apparent. One ml of 1% BSA-1×PBS with RNase inhibitor was added to the pellet, and the tube was flicked to mix. As the second washing step to remove all ACME, 1 ml of 1% BSA-1×PBS with RNase inhibitor was added to the tube, and the tubes were flicked. A 70 μm flowmi cell strainer was used to strain the cells to decrease the number of aggregates. The sample was centrifuged at 1500 *g* for 5 min at 4°C, and the cell pellet was visible.

The cells were checked on a hemocytometer by staining with Trypan Blue (1:1). The approximate number of cells was estimated, and their integrity was investigated. ACME-fixed cells were stained with the DNA dye DRAQ5 to sort intact single cells from cellular debris and aggregates (eBioscience 0.66 μl/ml of 5 mM stock). After staining, cells were incubated in the dark and on ice for 1 h. The stained cells were sorted using a BD FACSAria Fusion flow cytometer (BD Biosciences) with a red laser (640 nm). In total, 50,000 cells were sorted in collection buffer containing 1×PBS-1% BSA-RNAse inhibitor (40 U/ml) (Fig. S1). The cells were visualized under the far-red channel of a Nikon Ti2 Inverted fluorescence microscope (Fig. S1). The cells were preserved at −20°C by adding DMSO (10% final concentration).

### sc-RNA-seq via 10X Genomics and sequencing

Frozen cells were thawed on ice and centrifuged at 1500 *g* and 4°C. The supernatant was used to wash away the DMSO, followed by adding 1×PBS-1% BSA-RNase inhibitor (40 U/ml). Sorted cells were counted on a hemocytometer using Trypan Blue in a 1:1 ratio and diluted to yield ∼6000 cells for further processing. The cells underwent centrifugation at 1500 *g* at 4°C, then 10X Master Mix was added to the single cells. Subsequently, the cells were loaded onto a 10x Genomics Chromium chip. Gel bead emulsion (GEM) generation, barcoding, reverse transcription, cDNA amplification and library preparation were carried out according to the 10X Genomics protocol for 3′ Gene Expression (v3) user guide (10X Genomics CG000183 Rev C). The target cell recovery rate was set to 3000. After the GEM generation, a post-GEM-RT clean-up procedure was performed. cDNA amplification was performed for 16 cycles. The resulting cDNA was analyzed using a Qubit fluorometer and agarose gel electrophoresis. Samples were indexed for library preparation and library concentrations were quantified using KAPA PCR. The combined cDNA libraries were sequenced with a length of 150 bp on an Illumina NovaSeq 6000 sequencing platform at the Australian Genome Research Facility Ltd.

### Read alignment and cluster analysis

Quality checks of the library were performed using FastQC (Andrews, 2010) (Fig. S2). Raw sequencing reads were mapped to the reference *B. diegensis* genome using STARSolo (Kaminow et al., 2021 preprint). The genome file of *B. diegensis* (formerly *B. leachii*) was downloaded from the Aniseed Database (http://www.aniseed.fr) (Blanchoud et al., 2018). Gene models were identified using the StringTie software (Pertea et al., 2016). First, genome indices were created using STAR 2.7.9 (–genome SAindexNbases 12) (Dobin et al., 2013). The indices were then used to map the raw sequences to the *B. diegensis* genome using STAR 2.7.9, which masks the polyA tail during alignment; therefore, no prior trimming was performed. These run options were selected for the default barcode lengths using a droplet-type algorithm (soloUMIlen 12, soloType Droplet). No barcode read length defined (–soloBarcodeReadLength 0). The empty droplets were then filtered (soloCellFilter EmptyDrops_CR). Finally, sequences with barcodes present within the barcode whitelist were selected while mapping (soloCBwhitelist). The mapping statistics are listed in Table S1.

The sequencing dataset was evaluated for quality and saturation metrics to ensure its suitability for downstream analysis. A sequencing saturation value of 70.3% was achieved, indicating sufficient sequencing depth to capture the majority of transcripts within each cell. The dataset includes 13,621 detected genes, with an average of 1586 UMIs and 481 genes detected per cell, as reported in Table S1. These values are consistent with high-quality single-cell RNA sequencing datasets.

To further ensure robust clustering and marker gene analysis, we determined the optimal number of principal components (PCs) for dimensionality reduction. The elbow plot (Fig. S3) shows the standard deviation of PCs. This informed our choice to use 50 PCs for clustering and downstream analyses.

After mapping, clustering was performed using Seurat 4.0.1 (Stuart et al., 2019) and R 4.1.3. Before clustering, filtering was performed by selecting transcriptomes with 200-2000 genes expressed in at least three cells. The counts were normalized to the total counts using log normalization, and the scale factor was set to 10,000. Variability was identified within the 2000 genes using the *FindVariableFeatures* function with vst as the selection method. The linear dimensional reduction method was applied to the single-cell transcriptome using principal component analysis. The first 50 PCs for the mature colony were selected based on elbow and jackstraw analyses. Clustering was executed with a 0.8 resolution determined using the clustree R package (Zappia and Oshlack, 2018). The data were plotted using the nonlinear dimensional reduction method and presented using the Uniform Manifold Approximation and Projection (UMAP).

Doublet detection was performed using two independent methods: DoubletFinder and scDblFinder**.** The Seurat object containing 6500 cells was analyzed using each method to identify potential doublets. For DoubletFinder, parameter optimization was conducted using a parametric sweep to determine the optimal pK value, followed by an expected doublet rate of 1% per 1000 cells. Homotypic doublet proportions were estimated based on cluster annotations to adjust the number of expected doublets. For scDblFinder, default parameters were used, leveraging a statistical approach for doublet classification. The doublets identified by both methods were visualized on UMAP plots, confirming that they were scattered across clusters without dominating any specific cluster. This distribution suggests minimal impact on downstream analyses, including cluster marker identification and GO enrichment analysis. As such, all cells were retained for the final analysis.

Marker genes (top differentially expressed genes, log2FC>0.5, *P*<0.05) were identified for each cluster with Seurat's FindMarkers function (Table S1). Top genes were considered significant if *P*adj<0.05 and had a $\log_2$FC (fold change) value of >0.5 (Table S1). DEGs were annotated to their closest human orthologs. Genes that did not have vertebrate or human matches were also noted each cluster. The closest vertebrate orthologs that matched the DEGs were used to generate gene lists for GO analysis (Table S2). Finally, gene annotation was performed using Metascape (Zhou et al., 2019). The pathway and GO term enrichment were calculated by comparing them to the background genes (all the expressed genes in the dataset with an orthologue, i.e. 6039 genes). Clusters were annotated by integrating functional terms, known ascidian markers, and subsequent *in situ* hybridization.

To annotate immune-related clusters, we evaluated the expression of diagnostic markers supported by evidence in *Botrylloides* and related ascidian species (Franchi and Ballarin, 2014; Nicola and Loriano, 2017; Peronato et al., 2020; Clarke et al., 2024). Markers such as C3 (complement factor), Toll-like receptors (TLRs) and C-type lectins (CLECs) were selected based on prior functional characterization in tunicates or related organisms. For example, C3 has been previously associated with phagocytic cells, tunic amoebocytes and blood vessel epithelium (Clarke et al., 2024). Dot plot analysis was performed to assess the expression patterns of these and additional immune-related genes (e.g. CD209, CD63 and interferon regulatory factors), revealing enrichment in specific clusters (e.g. cluster 8 for CLEC genes, and clusters 9 and 19 for TLRs and IRFs) (Fig. S8).

Pseudo-time estimations of single cells were calculated using Monocle 3 (Trapnell et al., 2014), Scanpy 1.10.1 (Wolf et al., 2018), scVelo 0.3.2 (Bergen et al., 2020) and CellRank v. 2.0 (Lange et al., 2022). RaceID3/StemID2 0.3.9 was used to predict progenitor cell (pthr=0.01) (Herman et al., 2018). Code is available at https://github.com/MJWilsonOtago/scRNAseqBotrylloides/. Raw and processed data have been submitted to GEO under accession number GSE290754.

### Probe synthesis, *in situ* hybridization and image acquisition

Probe templates (∼500–700 bp) were PCR-amplified, cloned into pCRII-TOPO and confirmed by Sanger sequencing. Digoxygenin (DIG)-labeled probes were synthesized using SP6/T7 polymerases (Sigma-Aldrich). Primers were designed to amplify the genes of interest (Table S2). *In situ*

hybridisation was performed as described previously (Zondag et al., 2019). Images were acquired using a Nikon TiE with 40 or 60× magnification.

## Acknowledgements
We thank the Gemmell Lab of the University of Otago, particularly Joanne Gillum, for their contribution to the single-cell optimization. We also thank Bridget Fellows, Devon Gamble, Justine Gapuz and Ed Moody for their comments on the final draft. We thank the Australia Genome Research Facility for sequencing services.

## Competing interests
The authors declare no competing or financial interests.

## Author contributions
Conceptualization: M.J.W.; Data curation: B.T., M.M., M.J.W.; Formal analysis: B.T., M.M., M.J.W.; Funding acquisition: M.J.W.; Investigation: B.T.; Methodology: M.M., M.J.W.; Supervision: M.M., M.J.W.; Writing – original draft: B.T., M.J.W.; Writing – review & editing: M.M., M.J.W.

## Funding
This study was supported by funding from the Department of Anatomy and a University of Otago research grant. B.T. was supported by an Anatomy Department PhD scholarship from the University of Otago. Open Access funding provided by University of Otago. Deposited in PMC for immediate release.

## Data and resource availability
Code is available at https://github.com/MJWilsonOtago/scRNAseqBotrylloides/. Raw and processed data have been submitted to GEO under accession number GSE290754.

## Peer review history
The peer review history is available online at https://journals.biologists.com/dev/lookup/doi/10.1242/dev.204265.reviewer-comments.pdf

## Special Issue
This article is part of the Special Issue 'Lifelong Development: the Maintenance, Regeneration and Plasticity of Tissues', edited by Meritxell Huch and Mansi Srivastava. See related articles at https://journals.biologists.com/dev/issue/152/20.

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
