## [Peer Review File · Development (Cambridge, England)]

Single-cell transcriptomic profiling of the whole colony of *Botrylloides diegensis*: insights into tissue specialization and blastogenesis

Berivan Temiz, Michael Meier and Megan J. Wilson
DOI: 10.1242/dev.204265

Editor: James Briscoe

Review timeline

Original submission:	21 July 2024
Editorial decision:	26 October 2024
First revision received:	27 February 2025
Editorial decision:	27 March 2025
Second revision received:	13 May 2025
Accepted:	14 May 2025

Original submission

First decision letter

MS ID#: dev.204265

MS TITLE: Single-cell transcriptomic profiling of the whole colony of *Botrylloides diegensis*: Insights into tissue specialization and blastogenesis

AUTHORS: Berivan Temiz; Michael Meier; Megan J Wilson

Dear Dr Wilson,

I apologise that it has taken so long to review your study, we found it difficult to find referees. However, I am pleased to say we have now received all three referees' reports on the above manuscript, and have reached a decision. The referees' comments are appended below, or you can access them online: please go to:

As you will see, the referees express considerable interest in your work, but have some significant criticisms and recommend a substantial revision of your manuscript before we can consider publication. In my opinion there are three issues that need to be addressed. The absence of established markers such as *vasa* and *piwi* in the transcriptome analysis of blood cells seems to require more attention. Providing *in situ* hybridization validation for the budlet cluster (cluster 6) would strengthen the study. The code and dataset should be made available through appropriate public repositories. In addition, the referees suggest revisions that would increase clarity of the presentation of the Results. Please also attend to the comments on anatomical terminology and phylogeny.

If you are able to revise the manuscript along the lines suggested, which may involve further experiments, I will be happy to receive a revised version of the manuscript. Your revised paper will be re-reviewed by one or more of the original referees, and acceptance of your manuscript will depend on your addressing satisfactorily the reviewers' major concerns. Please also note that Development will normally permit only one round of major revision. If it would be helpful, you are welcome to contact us to discuss your revision in greater detail. Please send us a point-by-point

response indicating your plans for addressing the referees' comments, and we will look over this and provide further guidance.

Please attend to all of the reviewers' comments and ensure that you clearly highlight all changes made in the revised manuscript. Please avoid using 'Tracked changes' in Word files as these are lost in PDF conversion. I should be grateful if you would also provide a point-by-point response detailing how you have dealt with the points raised by the reviewers in the 'Response to Reviewers' box. If you do not agree with any of their criticisms or suggestions please explain clearly why this is so.

Reviewer 1

Temiz et al. use single-cell RNA sequencing (scRNAseq) to profile an entire colony of the colonial tunicate *Botrylloides diegensis*, previously misidentified as *Botrylloides leachii*. As far as I can tell, this is the first account of such single-cell profiling of an entire tunicate colony, in addition to the first single-cell profiling of at least one entire adult ascidian tunicate. These data advance the field in a substantial manner, and provide a rich dataset for the study of tunicates. I have some comments and suggested changes for the authors:

- 1) First, noting that the *B. diegensis* in this study was formerly identified as *B. leachii* should be at the very top of the manuscript. It was only deep into the methods that it was made clear that these were the same species across multiple publications.
- 2) Line 78: *Ciona intestinalis* should be replaced with just *Ciona*, as these studies covered a range of other species studied (e.g. *C. robusta*, *C. savignyi*).
- 3) Figure 1: legend says tunicate tree, but actual tree says "urochordata", while "ascidian" is used in the text. Some more consistency would be appreciated. In my opinion, "tunicate" should be preferred over the unaccepted junior synonym urochordata, and the paraphyletic ascidian.
- 4) Figure 2 seems to have been deleted in text, though a low-resolution copy appears at the end of the manuscript.
- 5) The authors should include an orthologue analysis and table for *Ciona*, *Styela*, and *Botryllus* genes. This would greatly increase the usefulness of the study for those studying other tunicate species.
- 6) I was disappointed to not see any Cluster 6 marker gene in situ hybridization images. This would seem to be important supporting evidence for the most interesting and unique cell cluster in this study (the bud disc).

Reviewer 2

SUMMARY OF THE ADVANCE MADE IN THIS PAPER AND ITS POTENTIAL SIGNIFICANCE TO THE FIELD

This manuscript is an RNA seq analysis of a colonial ascidian, and the authors show that they can identify specific adult tissues with the genes that were isolated by the technique. There are also 6-8 types of blood cells that have been described in colonial tunicates and some of these were identified, although it was not clear exactly which ones. The authors make a trajectory of developmental transitions based on their data, which is interesting and new.

SUGGESTIONS TO AUTHORS

This manuscript reviews single cell transcriptomic profiling of a colonial ascidian, *Botrylloides diegensis*. *B. diegensis* is known to go through cycles of renewal through the process of budding new zooids, and blastogenesis, the apoptosis of older zooids during the process of renewal. Colonial ascidians are also known to have several different stem cells circulating in their blood, so the results are important and relevant for publication in *Development*. Twenty-nine cell clusters were identified and those marking adult tissues, such as the gut and endostyle were confirmed by

in situ hybridization. The blood cell markers were less convincing, and the stem cell markers were not clearly defining stem cells. It is surprising that *vasa* and *piwi* were not found in the transcriptome analyses (line 307) because they have been used as markers in closely related colonial ascidians. There are some other important issues that should be addressed before publication, outlined below.

Major Issues

1. The phylogenetic tree in Figure 1 is misleading, as the position of *Oikopleura* is ever changing, and the Salps have been shown to be a sister group to the Phlebobranchia (Damian-Serrano et al. 2023). It might just be more accurate to put the Tunicata in place in the phylogeny instead.
2. Figure 1 - The correct term for the subphylum is Tunicata, not Urochordata. (Zeng and Swalla, 2005)
3. Line 419 - "As a more ancestral chordate, *B. diegensis* occupies an essential phylogenetic..." In fact, *B. diegensis* has been evolving for millions of years from the ancestral tunicate, so all we know is that *B. diegensis* is an invertebrate chordate, it is definitely not ancestral, as it is extant. One could also argue that these results are interesting, as *B. diegensis* is a colonial chordate, and those are found only in the tunicates.

References

1. A Damian-Serrano, M Hughes, K R Sutherland, A New Molecular Phylogeny of Salps (Tunicata: Thalicea: Salpida) and the Evolutionary History of Their Colonial Architecture, Integrative Organismal Biology, Volume 5, Issue 1, 2023, obad037, <https://doi.org/10.1093/iob/obad037>
2. L Zeng and B Swalla, Molecular Phylogeny of the Protochordates: Chordate Evolution, Canadian Journal of Zoology, 1(83), p. 24-33, 2005 DOI: 10.1139/z05-010

Reviewer 3

SUMMARY OF THE ADVANCE MADE IN THIS PAPER AND ITS POTENTIAL SIGNIFICANCE TO THE FIELD

Temiz and collaborators perform single cell transcriptomic analyses on a whole colony of the tunicate *Botrylloides diegensis*. The authors characterize clusters pertaining to the digestive system, blood cells, endostyle and bud primordium, most of the time validating the markers with in situ hybridization experiments.

The dataset here presented constitutes a valuable source of information for the tunicate community. This together with the mixture of techniques and methods to validate cell types in the colony, comprises a piece of work completed through a holistic approach that could be incredibly promising. The characterization at the single cell level of a colonial ascidian such as *B. diegensis* represents countless future experiments and comparisons at the evolutionary level and regeneration biology to say the least. However, there are serious concerns on how the data is presented and discussed, and some additional validations are needed.

SUGGESTIONS TO AUTHORS

The manuscript would benefit from a clearer figure presenting the umap and all the clusters, with the annotated names and cluster numbers stated clearly (also pointing out the not annotated clusters). It is not clear which clusters have been assigned and is confusing when looking for example at the PAGA analyses, in which the tSNE plot has a different topology. Other than that, UMAPs density plot are not enough to allow the reader to evaluate the expression of each marker, we strongly advise the authors to implement the figures with plots showing the expression of each gene for each individual cluster.

Authors should implement the manuscript with in situs of at least one of the marker expressed by the cells in the budlet cluster (cluster 6), to confirm the identity of this cluster.

Additionally, as in a colony coexist adult zooids, possessing differentiated tissues, and immature buds, that are still differentiating their organs, authors could show the expression of the same genes in different moments of organs differentiation. This could be achieved with the in situ they

already provided, showing bud and adults tissues. This could be also helpful for the genes they say are responsible for differentiation, that authors extract from the trajectory analyses.

The in situ hybridizations could be better presented, making clearer what kind of tissue is being observed, by adding more references in the pictures, and better description in the figures' captions, or schemes.

I would recommend the authors to make the code publicly available. It has not been possible to check the repository, and the code used for this project. Even though the data can be browsed in their shiny app, I would encourage the authors to make the final file downloadable so it can be browsed or used by the community (please do check cell browser USCSC or GEO).

The authors often refer to the endostyle as a stem cell niche. While it is true that the area near to this organ in *B. schlosseri* has been found hosting cells with hematopoietic-like expression pattern, evidences about the origin of these cells are scarce, but it is reported that progenitor/stem cells are found in the area ventral to the endostyle, not in the endostyle itself (see Rosental et al., 2018 DOI:10.1038/s41586-018-0783-x, Vanni et al., 2023 DOI: 10.1101/2023.05.15.540819).

As a general recommendation I encourage the authors to rename the sections of the manuscript. It is confusing to see a Results and Discussion and later on a Discussion section.

Results are full of methods that could stay in the methods section, to give more space to discussion, that sometimes is limited. For example, the section "Identification of Cluster Marker Genes" of the results could be entirely moved in the methods.

I would strongly suggest the authors to specify the species they are referring to, when citing a previous work on known markers (for example line 399-400, specify "in humans")

Some citations are missing (for example line 441-442 "Specialized progenitors associated with the vascular lining and epithelial tissues have also been identified.") or not totally fitting for the statement (for example, the endostyle description (line 196) cites a cyclostomata paper. Why not use a description available from papers describing the endostyle in tunicates?). And the same is true for diagnostic markers the author use to assign cluster identities, some markers are available from studies in botryllids (see for example Franchi and Ballarin 2017 DOI: 10.3389/fimmu.2017.00674, and citations in it for some markers of blood cells) and only rarely authors use such studies, available from closely related species.

An overall view of the atlas presented here is well discussed. However, is still lacking key discussion points based on the cluster annotations, this would increase the quality of the manuscript immensely. For example, the GO terms of the enriched biological processes and the relationship with the in situs in the body of the animal is superficially discussed. Moreover, I would like to see the authors discussing more about the quality and saturation of the sequencing. A simple elbow plot showing the saturation of sequenced UMIs would suffice. Later, the discussion dilutes towards the end into the caveats of the technique and the atlas itself instead of focusing on the actual results and how this improves the knowledge of cell types in colonial ascidians.

An overall view of the atlas presented here is well discussed. However, is still lacking key discussion points based on the cluster annotations, this would increase the quality of the manuscript immensely. For example, the GO terms of the enriched biological processes and the relationship with the in situs in the body of the animal is superficially discussed. Moreover, I would like to see the authors discussing more about the quality and saturation of the sequencing. A simple elbow plot showing the saturation of sequenced UMIs would suffice. Later, the discussion dilutes towards the end into the caveats of the technique and the atlas itself instead of focusing on the actual results and how this improves the knowledge of cell types in colonial ascidians.

Generally, I encourage the authors to better describe *B. diegensis* anatomy in the introduction section, instead of alphabetically listing the different organs, to make it clearer especially for people outside of the field. The asexual cycle could be also better described, better pointing out the staging system and better describing the stage that has been chosen for the single cell sequencing study.

Line 59 - The term "vein" for tunicates is not formally correct. While there are blood vessels (that have an anatomical structure that differs from the classical endothelium), they do not convey blood towards the heart, but everywhere, as they connect all the zooids, each of them having a heart. Authors should also mention that blood in zooids runs in sinuses, not separated by an endothelium (Burighel and Brunetti, 1971 <https://doi.org/10.1080/11250007109429158>).

Line 68 - I wouldn't list the bud as an anatomical structure of zooids, rather buds usually possess their own anatomical structures (and even other buds). As mentioned above, I would better describe the anatomy and the asexual cycle in the Introduction.

Line 71 - "Asexual reproduction and blastogenesis": aren't they the same thing? Or do authors mean something different? In the latter case it should be explained

Line 102 - The citation of the paper they refer to for the dissociation protocol is in the method section and figure legend but maybe it would be more useful for the reader to have it in the main text.

Figure 3. I find it difficult to locate and visualize the gene expression in the overall colony, maybe an extra graphic summarizing the in situs in a complete colony could help. Structures should be better indicated, and also the stage of the animals we are watching, are those all adults? Are they buds? Panel B is not well described both in the main text and in the figure caption, what is labeled in the panel with orange and black arrows? What is "bv"? If "bv" stands for blood vessel, and the image refers to a zooid section, it could be that the authors are instead pointing to epidermis.

Line 161 - "Tubb was broadly upregulated in branchial epithelial tissues of the digestive tract": by reading this I would suppose that it is expressed in the branchial epithelium, but it is expressed in different tissues. What do the authors mean by this?

-Figure 4. In B is not clear why the authors did not show the same representation of GO term enrichment as in a heatmap with all the rest of clusters

In supplementary file S3 the authors list also other galectin candidates they have found, that are also expressed in other clusters other than the ones they cite. How was the marker they show in figure 4 chosen? And which one is among the ones on the list? I would suggest the authors to better show the expression of each gene of these lists across all the clusters with a dotplot or a matrixplot, and this could be also applied to all the density plot they show. If authors implement in the figure panels with Dot plots for each UMAP, the expression in each cluster is clearer.

Line 274 - the text says Cluster 7 but figure 5C says Cluster 17

Also in figure 5, as in figure 4, the authors would benefit from better orientating the reader on what they are seeing, for example by adding more description of the structures around.

Figure 6 and Fig S4-S11. Why were the GO terms of this set of clusters not included in the main figure? Additionally, in the supplementary figures from the GO terms besides apoptosis, necroptosis is also enriched, could this mean that the cells were put under great stress and even going through apoptotic processes?

Figure 8. It is not clear which projection method the authors used for establishing the relationships based on PAGA calculations. Moreover, the choosing of tSNE projections in this figure is confusing, what was the criteria for picking this over UMAP projections?

How were markers identified in the genome? For example, soxb: in the *B. diegensis* genome available online, there are two annotated sox2/sox14/sox21. How did the authors choose the one they used? If this information comes from previous works that annotated such genes, it should be clearly mentioned.

Lines 295-296 - "we examined cells with thickened epithelium" - what do the authors mean by this?

Line 309 - "The proportion of dense cells was the same as that of bud disc genes" what do authors mean by this? If we are talking about cell proportions, I would expect to see a different graph from the umap, allowing the reader to clearly evaluate "cell proportions"

Line 353 - "This suggests that multiple common progenitor populations may give rise to differentiated cell types" - authors should better explain what suggests this and how.

-Line 369- "This plot was consistent with the PAGA plot, particularly with multiple transition events from clusters 0, 6,21, and 20 (Fig. 8A), suggesting that cells within these clusters serve as central nodes with multiple connections (Fig. 8D)." Do the authors suggest here for multipotency? If so, I recommend using extra bioinformatic tools to verify multipotency, like StemID (<https://doi.org/10.1016/j.stem.2016.05.010>)

-Line 425 - "In contrast, clusters not enriched for a particular group of markers or functional annotations may indicate an intermediate cell state." These cells could also represent doublets in the outcome. Also given that the pipeline followed by the authors did not use any doublet detection algorithm (i.e., Scrublet, Solo, LIGER, etc.) the possibility of doublets present in the final dataset is very likely and is not discussed in the manuscript.

-Line 448- "Adult stem cells (ASCs) are involved in various epithelial tissues in invertebrates. For example, in sponges, epithelial cells function similarly to stem cells and play a role in tissue repair." This is a major statement with only examples on sponges talking about all invertebrates. Are there any more examples so authors can generalize like this?

First revision

Author response to reviewers' comments

We sincerely thank you for your thoughtful and constructive feedback on our manuscript. Below, we address each of your comments and outline the revisions made to improve the manuscript. We have also included an annotated version of the manuscript, with sections in red to highlight the updated text.

Reviewer 1:

Comment: Noting that *B. diegensis* was previously identified as *B. leachii* should be clarified at the start of the manuscript.

Response: We have revised the introduction to explicitly state this clarification early on. Additionally, this information has been included in the methods section for completeness.

This has been added to the Introduction "*Botrylloides diegensis*, previously identified as *Botrylloides leachii* in earlier studies (Temiz et al., 2023), "

Comment: Replace "*Ciona intestinalis*" with "*Ciona*."

Response: We have updated "*Ciona intestinalis*" to "*Ciona species*" to reflect broader species inclusion.

Comment: Prefer "tunicate" over "urochordata."

Response: All instances of "urochordata" have been replaced with "Tunicata," in Figure 1 and and the figure has been updated to reflect more current relationships within this group (reviewer 2).

Comment: Ensure Figure 2 is properly included in the text.

Response: Figure 2 has been inserted again into the text. It seems to be a device/conversion

issue, as it was included in the Word document uploaded. This time, I inserted the figure as a different file type.

Comment: Include an orthologue table comparing genes across Ciona, Styela, and Botryllus.

Response: This information is already available using ANISEED (you can go to the gene cards and then link straight to the orthologous gene).

Comment: Include in situ hybridization for a marker gene from Cluster 6.

Response: These experiments were conducted, the new data has been incorporated into the manuscript (Figure 8).

Reviewer 2:

Comment: Revise the phylogenetic tree to reflect recent findings on Tunicata relationships.

Response: We updated the phylogenetic tree in Figure 1, based on recent molecular studies. The figure legend was revised to reflect this change.

Comment: Use "Tunicata" instead of "Urochordata."

Response: All references to "Urochordata" have been replaced with "Tunicata"

Comment: Avoid describing *B. diegensis* as ancestral.

Response: The text has been revised to "*As an extant colonial chordate*"

Comment: Address the lack of vasa and piwi in the transcriptome data.

Response: We discussed the absence of these markers in the context of technical limitations and potential biological factors. The absence of vasa and piwi markers may reflect technical limitations such as dropout events, which are common in single-cell RNA sequencing when detecting lowly expressed genes

Comment: Separate results and methods for improved clarity.

Response: We appreciate the suggestion to separate results from methods for clarity. Part of the results describes the outcomes of methodological steps, such as identifying cluster marker genes, which are essential to understanding the findings. For this reason, we retained some current structure. Much of the cluster marker paragraph has now been moved to the methods.

Comment: Provide clearer descriptions of endostyle and other anatomical structures.

Response: We have included more details regarding Botrylloides anatomy in the manuscript (see other comments below) and additional schema to aid interpretation of the in situ sections.

Reviewer 3:

Comment: The manuscript would benefit from a clearer figure presenting the umap and all the clusters, with the annotated names and cluster numbers stated clearly (also pointing out the not annotated clusters). It is not clear which clusters have been assigned, and it is confusing when looking, for example, at the PAGA analyses, in which the tSNE plot has a different topology. Other than that, UMAPs density plot are not enough to allow the reader to evaluate the expression of each marker, we strongly advise the authors to implement the figures with plots showing the expression of each gene for each individual cluster.

Response We appreciate the reviewer's detailed feedback regarding the visualization of cluster identities and gene expression. While we agree that clearly labeled cluster identities are

essential, we believe that the density plots we have used provide significant advantages, particularly for showing expression patterns.

Density plots allow readers to easily visualize clusters of cells with high expression intensity while also observing the distribution of expression in cells where the marker is present at a lower or more diffuse level. This is especially helpful in cases where expression is broad or specific subsets of cells within a cluster exhibit higher expression levels. In contrast, traditional dot plots or binary cluster marker assignments may obscure such nuanced patterns.

To address the reviewer's concerns, we believe that supplementing rather than replacing density plots will maintain the clarity and depth of analysis for both subsets and broadly expressed genes.

We hope this compromise effectively balances the clarity of cluster visualization with the interpretability of expression patterns across clusters.

Fig. 5 - We added cluster numbering to density plots and enlarged plots. - -

For Fig. 3, we added a schematic showing a whole zooid and where the sections are from (focus is on the gut), changed the individual violin plots to a stacked plot, to allow an increase in the size of the other panels.

Fig. 6 - We updated the expression plots to a more concise stacked format, enabling us to enlarge the in situ images. We also added a heatmap plot for GO and pathways.

Comment: Authors should implement the manuscript with in situs of at least one of the marker expressed by the cells in the budlet cluster (cluster 6), to confirm the identity of this cluster. Additionally, as in a colony coexist adult zooids, possessing differentiated tissues, and immature buds, that are still differentiating their organs, authors could show the expression of the same genes in different moments of organs differentiation.

Response:

We understand the reviewers' request for a developmental progression series to demonstrate changing expression intensities over time. However, several practical limitations make such experiments challenging within this study's scope. First, in situ hybridization is inherently non-quantitative and can be influenced by factors such as probe penetration efficiency and tissue-specific fixation. These variations make it difficult to quantify expression changes across developmental stages precisely.

The primary objective of our in situ experiments was to confirm the spatial expression of highly expressed cluster markers within specific cell types, validating our scRNA-seq analysis. This approach aligns with common practices in single-cell transcriptomic studies, where in situ validation serves to localize cell-type-specific markers rather than quantify expression dynamics over time. While a detailed expression series would be informative, achieving such a dataset would require significant additional resources and may not be feasible within the scope of this study.

We hope the reviewers understand the rationale behind our approach and appreciate that this study provides a valuable foundation for future work that could explore dynamic expression patterns in more depth.

Comment: The in situ hybridizations could be better presented, making clearer what kind of tissue is being observed, by adding more references in the pictures, and better description in the figures' captions, or schemes.

Response: We have added more schemes and identifiers to the images to aid with interpretation of the in situ results.

Comment : I would recommend the authors to make the code publicly available.

Response: We included a link to Github, which has the code used to generate the figures and the CellRank analysis. This has been updated to include more helpful notation.

Comment: The authors often refer to the endostyle as a stem cell niche. While it is true that the area near to this organ in *B. schlosseri* has been found hosting cells with hematopoietic-like expression pattern, evidences about the origin of these cells are scarce, but it is reported that progenitor/stem cells are found in the area ventral to the endostyle, not in the endostyle itself (see Rosental et al., 2018 DOI:10.1038/s41586-018-0783-x, Vanni et al., 2023 DOI: 10.1101/2023.05.15.540819). Clarify claims about the endostyle as a stem cell niche.

Response: We acknowledge that the distinction between a "niche" and a "source" of stem cells may need clarification. In our manuscript, we define the stem cell niche as a microenvironment that supports the maintenance, homing, and differentiation of stem cells into specialised cell types.

We have reviewed the references cited by the reviewer and included them in the revised manuscript as strong evidence supporting the role of the endostyle as a niche rather than a direct source of stem cells. Rosental et al. (2018) identified the ventral region near the endostyle as hosting hematopoietic-like stem cells, but also described the endostyle's role in supporting the differentiation of blood cells, aligning with the concept of a niche. Vanni et al. (2023) similarly discussed the ventral region as hosting progenitor cells while emphasising the supportive role of the endostyle in stem cell maintenance and differentiation. Jiang et al., 2024 Styela used spatial transcriptomics to characterise the cells located beside the endostyle (they called the hemolymphoid region) and showed multiple immune blood cell precursor cells.

We have revised the manuscript to clarify the distinction between a "niche" and a "source" and ensure that the terminology aligns with the established literature. The evidence indicates that while the endostyle may not be the primary source of hematopoietic stem cells, it provides a critical supportive environment for their differentiation and function.

Comment: not totally fitting for the statement (for example, the endostyle description (line 196) cites a Cyclostomata paper. Why not use a description available from papers describing the endostyle in tunicates?).

Response: Thank you for this comment. The original citation aimed to provide a general perspective on the function of the endostyle across species, including Cyclostomata. However, we understand the importance of focusing on tunicate-specific references for this context. We have revised the statement to include only citations relevant to tunicates, ensuring the description aligns more closely with the focus of the manuscript.

Comment: And the same is true for diagnostic markers the author use to assign cluster identities, some markers are available from studies in botryllids (see for example Franchi and Ballarin 2017 DOI: 10.3389/fimmu.2017.00674, and citations in it for some markers of blood cells) and only rarely authors use such studies, available from closely related species.

Response: Thank you for highlighting the importance of using diagnostic markers to assign cluster identities. We appreciate the reference to Franchi and Ballarin (2017) and have carefully reviewed their findings. Upon closer examination, we noted that many of the markers discussed in this study demonstrate variable expression across different ascidian species (e.g., C3) or are broadly categorized as being expressed by hemocytes without a clear association to specific cell types (e.g., "probably phagocytes"). This variability presents a challenge in directly applying these markers to identify distinct cell types or clusters in our dataset.

To ensure robustness in our analysis, we have prioritized markers with more consistent evidence or functional characterization in tunicates, particularly in *Botrylloides* or closely related ascidians. For example:

- C3 (Complement factor): Previously reported to be expressed in phagocytic cells, tunic amoebocytes, and blood vessel epithelium (Clarke et al., 2024). It is also upregulated following injury. While not a cluster marker, C3 is expressed in Cluster 8, consistent with

its role in phagocytic cells.

- Toll-like receptor (TLR): Associated with immune cell types such as morula and phagocytes in *Botryllus* (Peronator et al., 2020).

We also expanded our analysis to include additional immune-related genes (e.g., C-type lectins, CD209), using dot plots to identify expression patterns across clusters. We have included an additional paragraph in the results section.

Comment: An overall view of the atlas presented here is well discussed. However, is still lacking key discussion points based on the cluster annotations, this would increase the quality of the manuscript immensely. For example, the GO terms of the enriched biological processes and the relationship with the in situs in the body of the animal is superficially discussed.

Revision: Added more detailed discussion, although mindful of the word limit:

Discussion section - Insights from Gene Ontology and Pathway analysis

GO and pathway analyses of the top marker genes for each cluster provided a helpful starting point for functional annotation, revealing conserved gene regulatory programs in *B. diegensis*. For instance, Clusters 9 and 19 showed significant enrichment for terms related to phagocytic vesicles, lysosomes, and immunological synapses—consistent with a hemocyte-like lineage. Similarly, Clusters 10 and 11 were enriched for ciliary processes, corroborating our in situ hybridization results indicating that *Tubb* and *Cnfn* are expressed in highly ciliated epithelia. Despite these insights, GO analysis has limitations. First, many annotations rely on the “closest” vertebrate or human ortholog, an approach that may overlook species-specific roles or misinterpret homology in tunicates. Second, if a cluster primarily comprises transient or intermediate cell states, its marker genes may not fall into neat functional categories. As a result, certain clusters—particularly those with broadly expressed transcripts—did not yield strong enrichments for any one function. These patterns echo previous studies in other colonial ascidians that describe similarly “ambiguous” blood cell types or unannotated genes with limited homology to model organisms (Rosental et al., 2018). Thus, while GO analysis can pinpoint broad functional themes, it must be interpreted cautiously, especially in organisms where many genes remain uncharacterized and

dynamic and transitory cell populations are likely abundant.

Comment: I would like to see the authors discussing more about the quality and saturation of the sequencing. A simple elbow plot showing the saturation of sequenced UMIs would suffice

Response: We thank the reviewer for highlighting the importance of discussing sequencing saturation and data quality in greater detail. In response to the comment, we have included additional information regarding the sequencing quality and saturation metrics in the manuscript. Specifically, we highlight the sequencing saturation value of 70.3% and the mean UMI per cell of 1586, as reported in Supplementary Table S1. These metrics indicate that the sequencing depth was sufficient to capture the majority of transcripts within each cell while maintaining high data quality.

Furthermore, we have now provided an elbow plot (Supplementary Figure S3) illustrating the distribution of principal components to clarify the choice of 50 PCs for downstream analysis. Using PC50 is a standard approach, especially when the elbow plot does not clearly plateau, as it allows the inclusion of significant variance contributions without overfitting. While the elbow plot is primarily used to determine the optimal number of PCs, the sequencing saturation metric and the UMI statistics directly address the sequencing depth and quality. Together, these demonstrate that the dataset is high-quality and adequately powered for the analyses presented.

Comment: As a general recommendation I encourage the authors to rename the sections of the manuscript. Rename sections and move methods-heavy parts of results.

Response: This has been corrected - to “Results” Rename sections for clarity and removed some details to the methods section.

Comment: , I encourage the authors to better describe *B. diegensis* anatomy in the introduction section

Response: We expanded the introduction to provide more of an overview of anatomy and blastogenesis while being conscious of the manuscript’s word limit and the need to keep this concise.

We have reorganized such that the budding schematic that was in a later figure is now part of Fig1 so that it is presented earlier on with the introduction material. We have expanded the description of the asexual cycle. To complement the updated Figure 1 to provide more background on how new zooids develop to help readers understand the stages depicted in panels F and G. We have previously published more detailed morphological descriptions, including blood flow in the colony.

Comment: Line 59 - The term “vein” for tunicates is not formally correct. Authors should also mention that blood in zooids runs in sinuses, not separated by an endothelium (Burighel and Brunetti, 1971 <https://doi.org/10.1080/11250007109429158>).

Response: This has been updated to “Blood circulates throughout the colony via a network of vessels embedded in the tunic, connecting with the zooid and bud sinuses. Each zooid has its own heart-driving circulation within interconnected vessels (Burighel and Brunetti, 1971; Mukai et al., 1978).” “vein” has been replaced with vessels

Comment: Line 68 - I wouldn’t list the bud as an anatomical structure of zooids, rather buds usually possess their own anatomical structures (and even other buds).

Response: This reference has been removed

Comment: Line 71 - “Asexual reproduction and blastogenesis”: aren’t they the same thing? Or do authors mean something different? In the latter case it should be explained

Response: This has been corrected.

Comment: Line 102 - The of the paper they refer to for the dissociation protocol is in the method section and figure legend but maybe it would be more useful for the reader to have it in the main text.

Response: This has been added to the main text

Comment: Figure 3. I find it difficult to locate and visualize the gene expression in the overall colony, maybe an extra graphic summarizing the in situs in a complete colony could help. Structures should be better indicated, and also the stage of the animals we are watching, are those all adults? Are they buds? Panel B is not well described both in the main text and in the figure caption, what is labeled in the panel with orange and black arrows? What is "bv"? If "bv" stands for blood vessel, and the image refers to a zooid section, it could be that the authors are instead pointing to the epidermis.

Response: We revised Figures 3 and 5 to include annotated UMAPs and dot plots, and improved figure legends for clarity.

Comment: Clarify UMAP annotations, cluster names, and include plots for marker gene expression.

Response: We updated Figures to include annotated UMAPs with cluster names and numbers.

Comment: Line 161 - "Tubb was broadly upregulated in branchial epithelial tissues of the digestive tract": by reading this, I would suppose that it is expressed in the branchial epithelium; what do the authors mean by this?

Response: This has been clarified in the text to "*Tubb* was upregulated in cells found in the epithelial tissues of the digestive tract, endostyle, and stigmata (Fig. 3D)."

Comment: Line 274 - the text says Cluster 7 but figure 5C says Cluster 17

Response: This was Fig. 6C/line 334 - this has been corrected in the text.

Comment: Figure 4. In B is not clear why the authors did not show the same representation of GO term enrichment as in a heatmap with all the rest of clusters

Response: A heatmap GO enrichment has been added to Fig. 6

Comment: In supplementary file S3 the authors list also other galectin candidates they have found, that are also expressed in other clusters other than the ones they cite. How was the marker they show in Figure 4 chosen? And which one is among the ones on the list? I would suggest the authors to better show the expression of each gene of these lists across all the clusters with a dot plot or a matrix plot, and this could also be applied to all the density plots they show.

Response: We appreciate the reviewer's suggestion to include a comprehensive dot plot to clarify the expression of galectin candidates across clusters. In response, we have provided a dot plot (Figure S5), incorporating all the candidate genes linked to the endostyle in other ascidians (from Supp. File S3).

We also added a dot plot for the candidate endostyle transcripts, Figure S7 including all L-galectin genes. Six out of the seven candidate genes (shown in Figure 5) show a clear enrichment in Cluster 3, supporting our prediction that this cluster represents the endostyle. These findings align with our initial interpretation in the manuscript. The seventh gene, while not exclusively enriched in Cluster 3, shows expression in this cluster and multiple other clusters. This broader expression pattern likely reflects its role in more general biological processes, rather than being endostyle-specific. These results strengthen our hypothesis by consistently clustering candidate genes within Cluster 3. We have updated the manuscript to incorporate this additional evidence and to discuss the broader expression of the seventh gene

We have revised the text in the manuscript to elaborate on the rationale behind selecting markers for detailed analysis, emphasising their biological significance and specific expression patterns. We performed an updated analysis incorporating all identified candidate genes that may be orthologous to those linked to the endostyle in other ascidians. Of these, six genes show specific enrichment in Cluster 3, further supporting our prediction that this cluster represents the

endostyle. One gene, a *Hox* gene orthologue, exhibits broader expression across multiple clusters. Figures have been added to the supplementary data.

While we acknowledge the reviewer's suggestion to apply dot plots to all density plots, we opted to retain the density plots for e.g., easier visualizing levels of expression across UMAP, which provides complementary insights to the dot plots.

Comment: Figure 6 and Fig S4-S11. Why were the GO terms of this set of clusters not included in the main figure? Additionally, in the supplementary figures from the GO terms besides apoptosis, necroptosis is also enriched, could this mean that the cells were put under great stress and even going through apoptotic processes?

Response: The exclusion of GO terms from the main figure was primarily due to space constraints. Including these graphs alongside the in situ results would have overcrowded the page, making it difficult to visualize and interpret the data effectively. However, these GO terms are detailed in the supplementary figures to ensure accessibility. The colony undergoes a weekly turnover of zooids, so typically, apoptotic cells are present or cells are about to undergo apoptosis. We did reduce the chance of this by FACS sorting and removing dead cells.

Comment: Figure 8. It is not clear which projection method the authors used for establishing the relationships based on PAGA calculations. Moreover, the choosing of tSNE projections in this figure is confusing, what was the criteria for picking this over UMAP projections?

Response: The tSNE projections made it easier to see the predicted trajectories with velocity arrows plotted.

Comment: How were markers identified in the genome? For example, *soxb*: in the *B. diegensis* genome available online, there are two annotated *sox2/sox14/sox21*. How did the authors choose the one they used? If this information comes from previous works that annotated such genes, it should be clearly mentioned.

Response: We used previously identified genes (see references provided), and we have characterised *Sox* members through phylogenetics (<https://doi.org/10.1016/j.j.gen.2015.09.013>). This gene was the closest orthologue to *SoxB1*.

Comment: Lines 295-296 - "we examined cells with thickened epithelium" - what do the authors mean by this? Line 309 - "The proportion of dense cells was the same as that of bud disc genes" what do authors mean by this? If we are talking about cell proportions, I would expect to see a different graph from the umap, allowing the reader to evaluate "cell proportions clearly"

Response: These lines have been cut/edited to prevent clarify - "*To identify the cells forming the bud disc of the peribranchial epithelium, we focused on identifying the cells with the highest expression of transcription factors found in disc cells,*"

Line 353 - "This suggests that multiple common progenitor populations may give rise to differentiated cell types" - authors should better explain what suggests this and how.

Response: We have clarified this in the text. We observe that certain clusters in the PAGA graph (e.g., clusters 20, 21, 0, and 6) occupy central nodes with multiple outgoing connections, indicating high transcriptional connectivity to numerous other clusters. In lineage inference, this pattern typically reflects progenitor-like states that can branch toward different differentiated cell fates. Specifically, edges in PAGA represent a probabilistic measure of lineage transitions based on transcriptional similarity and RNA velocity; hence, nodes with multiple strong edges are interpreted as common progenitors with the potential to give rise to various terminal clusters. This interpretation is further supported by the velocity plots, which indicate the direction of transcriptional change emanating from these central nodes, reinforcing the idea that multiple progenitor populations lead to distinct, specialized cell types.

Comment: "In contrast, clusters not enriched for a particular group of markers or functional

annotations may indicate an intermediate cell state." These cells could also represent doublets in the outcome.

Response: We appreciate the reviewer's concern regarding the potential presence of doublets and the possibility of these affecting clusters not enriched for specific markers or functional annotations. To address this, we applied two independent doublet detection algorithms, DoubletFinder and scDblFinder, to identify potential doublets in our dataset. Both methods used parameter optimization to account for dataset-specific properties, including expected doublet rates for 10x sequencing (1% per 1,000 cells) and homotypic doublet proportions.

The results from these methods showed consistent classifications of doublets, which were distributed throughout the clusters rather than concentrated in specific ones. Visualisation of the detected doublets on UMAP plots confirmed their scattered distribution, suggesting they do not contribute to any particular cluster. Furthermore, excluding doublets had minimal impact on downstream analyses, including cluster marker identification and GO enrichment. Therefore, while doublets were identified, their presence does not influence the key biological interpretations of our study. This has been added to the methods.

Comment: Line 448- "Adult stem cells (ASCs) are involved in various epithelial tissues in invertebrates. For example, in sponges, epithelial cells function similarly to stem cells and play a role in tissue repair." This is a major statement with only examples on sponges talking about all invertebrates. Are there any more examples so authors can generalize like this?

Response:

We kept this brief due to the word limit but now have expanded this section in the discussion. There are numerous examples of adult stem cells in invertebrates - Adult stem cells (ASCs) contribute to various epithelial tissues and play critical roles in tissue regeneration and maintenance across invertebrates. For instance, epithelial cells in sponges function similarly to stem cells in tissue repair (Rinkevich et al., 2022). Abundant examples in other invertebrates highlight the role of ASCs in whole-organism processes, such as asexual reproduction and regeneration. In colonial ascidians, we propose that progenitor cells located within Cluster 6 contribute to the development and maintenance of the entire zooid, analogous to similar roles observed in other epithelial progenitor systems

Comment: Line 369- "This plot was consistent with the PAGA plot, particularly with multiple transition events from clusters 0, 6, 21, and 20 (Fig. 8A), suggesting that cells within these clusters serve as central nodes with multiple connections (Fig. 8D)." Do the authors suggest here for multipotency? If so, I recommend using extra bioinformatic tools to verify multipotency, like StemID (<https://doi.org/10.1016/j.stem.2016.05.010>)

Response: Thank you for the suggestion to utilize StemID for verifying multipotency. We have conducted additional analyses using StemID/RaceID, which further support the conclusions of the analyses in the submitted manuscript. However, we have decided not to include these results in the main text for the following reasons: RaceID re-clusters cells, which results in new cluster IDs that are inconsistent with the Seurat clusters referenced throughout the manuscript. This discrepancy makes it challenging to directly map the results to the original clusters described in our analyses, potentially leading to confusion. Unlike CellRank or scVelo, StemID does not incorporate velocity models for understanding transitions between transcriptional states, which were a central component of our trajectory analyses. Therefore, while StemID provides additional insights, it is less aligned with the primary objectives of our study.

For transparency, we have included the results from the StemID analysis below.

StemID Lineage Analysis - Bar plot showing the number of links and delta entropy for each cluster, highlighting the most probable multipotent clusters.

Seurat cluster 6 genes primarily align with StemID cluster 5, which is identified as a multipotent cluster based on the number of links and entropy metrics.

StemID-derived clusters. Note that the re-clustering by StemID results in different cluster IDs compared to the Seurat clusters used in the main manuscript

Burighel, P., Brunetti, R., 1971. The Circulatory System in the Blastozoid of the Colonial Ascidian *Botryllus Schlosseri* (Pallas). *Bollettino di zoologia* 38, 273-289.

Mukai, H., Sugimoto, K., Taneda, Y., 1978. Comparative studies on the circulatory system of the compound ascidians, *Botryllus*, *Botrylloides* and *Symplegma*. *J Morphol* 157, 49-77.

Rosental, B., Kowarsky, M., Seita, J., Corey, D.M., Ishizuka, K.J., Palmeri, K.J., Chen, S.-Y., Sinha, R., Okamoto, J., Mantalas, G., 2018. Complex mammalian-like haematopoietic system found in a colonial chordate. *Nature* 564, 425-429.

Temiz, B., Clarke, R.M., Page, M., Lamare, M., Wilson, M.J., 2023. Identification and characterisation of *Botrylloides* (Styelidae) species from Aotearoa New Zealand coasts. *New Zealand Journal of Marine and Freshwater Research*, 1-19.

Second decision letter

MS ID#: dev.204265R1

MS TITLE: Single-cell transcriptomic profiling of the whole colony of *Botrylloides diegensis*: Insights into tissue specialization and blastogenesis

AUTHORS: Berivan Temiz; Michael Meier; Megan J Wilson

Dear Dr Wilson,

I have now received all the referees reports on the above manuscript, and have reached a decision. The referees' comments are appended below, or you can access them online: please go to .

The overall evaluation is positive and we would like to publish a revised manuscript in Development, provided that the referees' comments can be satisfactorily addressed. Referee 3 has some constructive comments and requests for clarification that would be helpful to address. Please attend to this reviewer's comments in your revised manuscript and detail them in your point-by-point response. If you do not agree with any of their criticisms or suggestions explain clearly why

this is so. If it would be helpful, you are welcome to contact us to discuss your revision in greater detail. Please send us a point-by-point response indicating your plans for addressing the referees' comments, and we will look over this and provide further guidance.

Reviewer 1

The authors appear to have addressed all my previous concerns!

Reviewer 2

This manuscript reviews single cell transcriptomic profiling of a colonial ascidian, *Botrylloides diegensis*. *B. diegensis* is known to go through cycles of renewal through the process of budding new zooids, and blastogenesis, the apoptosis of older zooids during the process of renewal. Colonial ascidians are also known to have several different stem cells circulating in their blood, so the results are important and relevant for publication in *Development*. Twenty-nine cell clusters were identified and those marking adult tissues, such as the gut and endostyle were confirmed by in situ hybridization. The revision expands the description of the blood cell markers and suggest developmental trajectories where possible.

Many of the concerns of the reviewers have been addressed in this revised manuscript, thank you to the authors!

Reviewer 3

SUMMARY OF THE ADVANCE MADE IN THIS PAPER AND ITS POTENTIAL SIGNIFICANCE TO THE FIELD

Temiz et al., present the first single cell atlas of a colonial tunicate, *Botrylloides diegensis*. The database presented is potentially a great resource for the community and for future evolutionary comparative analyses.

SUGGESTIONS TO AUTHORS

The adjustments made by the authors to figures and text improved the general clarity of the manuscript.

I thank the authors for including pictures of the different colony parts for the markers of cluster 6. I agree with the authors view regarding quantification of gene expression by in situ.

From the newly included in situs, and from the authors description, not only the budlet primordium express the two genes of interest (*Col24a1* and *Lgal4/7*) but also cells found in blood vessels.

Therefore, I agree with the author's claims to cluster 6 identity, discussing the possibility that this cluster represents a population of pluripotent cells that can derive both from the peribranchial epithelium and/or from stem-like blood cells lining the vascular epithelium. However, i would like to point out to authors that "cells lining the blood vessel epithelium" (the same found in Rinkevich et al. 2010) are not described in the original finding as epithelial cells. Hence, the identity of progenitor-like cells as "epithelial" population should be clarified and better discussed.

Is it also to note that trans- and de-differentiation processes have never been demonstrated in the blastogenesis of colonial ascidians. Surely, dedifferentiation or transdifferentiation are not mentioned in the papers cited by the authors (Manni and Burighel, 2006; Manni et al., 2007, line 522).

I would like to encourage the authors to make available also the processed dataset, this is a valuable piece of knowledge that could be used in comparative evolutionary studies.

It is appreciated the authors included StemID and RaceID analyses after the first round of reviews. They argue RaceID re-clusters the cells, and this is true. However, there are way around this issue where the cell embeddings can be transferred from the object in R to the input of the package. By seeing these results on the UMAP projection, even in a supplementary file, it would validate further author's claim on the respective trajectories.

tSNEs are debated to not show accurately the cell transcriptional landscape of a given sample (Marx, 2024 <https://doi.org/10.1038/s41592-024-02301-x>).

Line 51. References are needed for previous single cell atlases

Line 70. Change "cell type" accordingly to either cell broad type or cell categories

Line 97. Rephrase or change the term 'separated'. This might be misunderstood with 'dissociated', 'annotated' or 'characterised'.

Line 159. Gene g14532 (referred as cornifelin) is different to what shown in Figure 3C. If Figure 3F is supposed to be referred here, the gene name is different, are these two different genes? please clarify.

Line 280. Gene g13310 is in Figure 6D not C. Description of expression of g11109 was skipped. Please label this figure appropriately and add proper descriptions of all genes shown.

Line 375-376. A large nucleus does not reflect a stem-like cell (what "large" means?), other authors refer to stem-like cells as having a high nucleus/cytoplasm ratio, maybe this was what the author meant?

Line 544. Ampullae niches have been described only in Vanni et al., 2022, not cited in the text.

Figure 2. I recommend using the same colour labelling for the UMAP in B and the heatmap in C.

Furthermore, in C it would be easier for the reader to add an annotation of each gene, or the name assigned next to each.

Figure 4. A. More explanation is needed in the legend or in the text. What are the edges in the graph? Does it tell more to the reader than the barplots in the supplementary?

B. Why not showing a heatmap of all the enriched terms in all clusters? It would be equally informative as A. and less ambiguous, if the goal is to show the biological processes enriched in the whole dataset.

Figure 5. In E, the expression of *Tubb* is redundant as it was already shown in Figure 3C.

Figure 6. F. Why in this figure the GO terms are showed as heatmap and in the previous one as a barplot? As a minor commentary I suggest a standard way of showing GO term results for all figures

Figure 10. D. It is not clear which terminal state corresponds to which cluster in the umap. I would recommend to colour code the same as B in each example of trajectory gene expression

Second revision

Author response to reviewers' comments

Dear Prof. Briscoe,

Thank you for the opportunity to revise our manuscript for *Development*. We appreciate the constructive feedback from Reviewer 3 and have carefully addressed all comments and suggestions, including incorporating additional analyses using the StemID/RaceID pipeline while retaining the original Seurat-based clustering.

Please find enclosed our revised manuscript and a point-by-point response detailing how we addressed each of the reviewer's comments.

We appreciate your consideration and look forward to your feedback.

A/P Megan Wilson

Reviewer 1: The authors appear to have addressed all my previous concerns!

Reviewer 2: This manuscript reviews single cell transcriptomic profiling of a colonial ascidian, *Botrylloides diegensis*. *B. diegensis* is known to go through cycles of renewal through the process of budding new zooids, and blastogenesis, the apoptosis of older zooids during the process of renewal. Colonial ascidians are also known to have several different stem cells circulating in their blood, so the results are important and relevant for publication in *Development*. Twenty-nine cell clusters were identified and those marking adult tissues, such as the gut and endostyle were confirmed by in situ hybridization. The revision expands the description of the blood cell markers and suggest developmental trajectories where possible.

Many of the concerns of the reviewers have been addressed in this revised manuscript, thank you to

the authors!

Reviewer 3: SUMMARY OF THE ADVANCE MADE IN THIS PAPER AND ITS POTENTIAL SIGNIFICANCE TO THE FIELD

Temiz et al., present the first single cell atlas of a colonial tunicate, *Botrylloides diegensis*. The database presented is potentially a great resource for the community and for future evolutionary comparative analyses.

SUGGESTIONS TO AUTHORS

The adjustments made by the authors to figures and text improved the general clarity of the manuscript.

I thank the authors for including pictures of the different colony parts for the markers of cluster 6. I agree with the authors view regarding quantification of gene expression by in situ. From the newly included in situs, and from the authors description, not only the budlet primordium express the two genes of interest (*Col24a1* and *Lgal4/7*) but also cells found in blood vessels. Therefore, I agree with the author's claims to cluster 6 identity, discussing the possibility that this cluster represents a population of pluripotent cells that can derive both from the peribranchial epithelium and/or from stem-like blood cells lining the vascular epithelium.

However, i would like to point out to authors that "cells lining the blood vessel epithelium" (the same found in Rinkevich et al. 2010) are not described in the original finding as epithelial cells. Hence, the identity of progenitor-like cells as "epithelial" population should be clarified and better discussed.

Response: We thank the reviewer for this clarification. We agree that Rinkevich et al. (2010) do not describe the blood vessel-associated progenitor cells as epithelial per se. Our intent was not to equate these cells with a canonical epithelial identity, but rather to acknowledge the range of proposed stem/progenitor cell locations across studies in this section of the discussion – including vascular linings, ampullae, and epithelial structures such as the endostyle. We have clarified this in the text.

Is it also to note that trans- and de-differentiation processes have never been demonstrated in the blastogenesis of colonial ascidians. Surely, dedifferentiation or transdifferentiation are not mentioned in the papers cited by the authors (Manni and Burighel, 2006; Manni et al., 2007, line 522).

Response: We thank the reviewer for this clarification. We agree that dedifferentiation and transdifferentiation have not been conclusively demonstrated in colonial ascidians such as *Botrylloides diegensis* or *Botryllus schlosseri*. The references to Manni and colleagues were meant to highlight the proposed multipotency of the peribranchial epithelium rather than to imply any direct evidence of dedifferentiation.

To address this point, we have revised the manuscript to: Clarify that the cited studies do not claim dedifferentiation or transdifferentiation, emphasize that such processes have been described only in a few budding ascidians (e.g., *Polyandrocarpa misakiensis*), and frame our interpretation of Cluster 6 in a more cautious manner.

I would like to encourage the authors to make available also the processed dataset, this is a valuable piece of knowledge that could be used in comparative evolutionary studies.

Response: Yes, both the processed and raw data have been uploaded to NCBI. The accession number has been included in the manuscript.

It is appreciated the authors included StemID and RaceID analyses after the first round of reviews. They argue RaceID re-clusters the cells, and this is true. However, there are way around this issue where the cell embeddings can be transferred from the object in R to the input of the package. By seeing these results on the UMAP projection, even in a supplementary file, it would validate further author's claim on the respective trajectories.

Response: To strengthen our identification of progenitor cells, we performed StemID2 analysis using

Seurat cluster IDs. Please note that RaceID internally reindexes clusters starting from 1, so Seurat cluster 6 is referred to as cluster 7 in the output. This cluster demonstrated the highest number of significant links and the highest StemID score (combining entropy and lineage connectivity), consistent with a multipotent progenitor identity.

Line 51. References are needed for previous single cell atlases

Response: References have been added.

Line 70. Change "cell type" accordingly to either cell broad type or cell categories

Response: This has been changed to categories.

Line 97. Rephrase or change the term 'separated'. This might be misunderstood with 'dissociated', 'annotated' or 'characterised'.

Response: This has been changed to "resolved" to clarify resolution by clustering.

Line 159. Gene g14532 (referred as cornifelin) is different to what shown in Figure 3C. If Figure 3F is supposed to be referred here, the gene name is different, are these two different genes? please clarify.

Response: Sorry, the gene number was incorrect in the main text. This has been corrected.

Line 280. Gene g13310 is in Figure 6D not C. Description of expression of g11109 was skipped. Please label this figure appropriately and add proper descriptions of all genes shown.

Response: The description of g11109 (Gst3a) was included in the text. However, the panel letters hadn't been updated to account for the inclusion of the new A panel. Apologies, this has been corrected now.

Line 375-376. A large nucleus does not reflect a stem-like cell (what "large" means?), other authors refer to stem-like cells as having a high nucleus/cytoplasm ratio, maybe this was what the author meant?

Response: Yes, a large nucleus, little cytoplasm. This has been clarified in the text to "Blood cells bordering the vascular epithelium (Fig. 8E vi-viii, orange arrowheads) appear stem-like, with a large nucleus to cytoplasm ratio (Fig. 8E vi insert)."

Line 544. Ampullae niches have been described only in Vanni et al., 2022, not cited in the text.

Response: We have included the following reference: Vanni, V., et al. (2023). "New permanent stem cell niche for development and regeneration in a chordate." [bioRxiv: 2023.2005.2015.540819](https://doi.org/10.1101/2023.2005.2015.540819).

Figure 2. I recommend using the same colour labelling for the UMAP in B and the heatmap in C. Furthermore, in C it would be easier for the reader to add an annotation of each gene, or the name assigned next to each.

Response: We have updated the colours on the heatmap. As many of these do not have an orthologue or name as yet, however their geneID is unlikely to change, we have left it as it is.

Figure 4. A. More explanation is needed in the legend or in the text. What are the edges in the graph? Does it tell more to the reader than the barplots in the supplementary?
B. Why not showing a heatmap of all the enriched terms in all clusters? It would be equally informative as A. and less ambiguous, if the goal is to show the biological processes enriched in the whole dataset.

Response: We appreciate the reviewer's feedback and have revised both the legend and main text to clarify the function of the network in Figure 4A. The edges in the Metascape network represent functional similarity between GO terms based on shared genes or semantic similarity. This layout groups related biological processes into functional modules, helping

the reader interpret broader biological themes across multiple clusters. This complements the barplots in the supplementary figure, which are less integrated.

Regarding Figure 4B, we chose to display a focused subset of representative clusters relevant to digestive tissue identity to highlight distinct biological processes across these tissue types. Including all clusters would dilute this message and introduce terms unrelated to digestive function (e.g., from neural or vascular clusters). The full list of GO terms are available in the supplementary files.

Updated legend: Figure 4. GO and pathway analyses for cell Clusters associated with digestive functions. A. Network plot of enriched biological processes from Metascape analysis. Each node represents a GO or pathway term, and edges indicate functional similarity based on shared gene content or annotation. Nodes are grouped and colored by cluster assignment and predicted tissue type. This network illustrates how processes such as lysosomal degradation, mitochondrial metabolism, and ciliary function cluster into broader biological themes across digestive-associated cell types. B. Heatmap showing $-\log_{10}(p\text{-value})$ of selected enriched GO terms in clusters associated with the digestive tract and stigmata. Clustering was performed on both GO terms and cell clusters to highlight distinct biological functions. Each row represents an enriched GO term, and each column represents the gene list for Clusters 14, 4, 5, 10 and 11

Figure 5. In E, the expression of Tubb is redundant as it was already shown in Figure 3C.

Response: We are highlighting the expression of Tubb in the endostyle here (whereas Fig. 3C was focused on the digestive tract), with a higher magnification image.

Figure 6. F. Why in this figure the GO terms are showed as heatmap and in the previous one as a barplot? As a minor commentary I suggest a standard way of showing GO term results for all figures

Response:

We thank the reviewer for pointing this out. The choice of GO term visualization was tailored to the purpose of each figure: In Figure 4, the focus is on highlighting biological processes specifically associated with the endostyle and digestive tract, where a network-based layout and barplots emphasize functional clustering and key pathways relevant to gut physiology. This presentation is effective for summarizing distinct tissue identities and conveying semantic relationships between terms.

In contrast, Figure 6F aims to compare and contrast the functional profiles of several blood cell clusters (Clusters 7, 8, 13, 17, and 25). For this purpose, a heatmap visualization was selected to allow direct comparison of term enrichment across clusters. This format enables readers to identify shared or distinct biological features among candidate hematopoietic populations. The legend has been updated to clarify this.

Figure 10. D. It is not clear which terminal state corresponds to which cluster in the umap. I would recommend to colour code the same as B in each example of trajectory gene expression

Response: Thank you for the suggestion. We have colour coded the names at the top of each heatmap to match the UMAP and labelled these terminal state clusters on the UMAP

Third decision letter

MS ID#: dev.204265R2

MS TITLE: Single-cell transcriptomic profiling of the whole colony of *Botrylloides diegensis*: Insights into tissue specialization and blastogenesis

AUTHORS: Berivan Temiz; Michael Meier; Megan J Wilson

ARTICLE TYPE: Research Article

Dear Dr Wilson,

I am happy to tell you that your manuscript has been accepted for publication in Development, pending our standard publication integrity checks.